

# Genomic insights into CKX genes: key players in cotton fibre development and abiotic stress responses

Rasmieh Hamid[1], Feba Jacob[2], Zahra Ghorbanzadeh[3], Mojtaba Khayam Nekouei[4], Mehrshad Zeinalabedini[3], Mohsen Mardi[3], Akram Sadeghi[5], Sushil Kumar[6] and Mohammad Reza Ghaffari[3]

[1] Department of Plant Breeding, Cotton Research Institute of Iran (CRII), Agricultural Research, Education and Extension Organization (AREEO), Gorgan, Golestan, Iran
[2] Centre for Plant Biotechnology and Molecular Biology, Kerala Agricultural University, Thrissur, Kerala, India
[3] Department of Systems Biology, Agricultural Biotechnology Research Institute of Iran (ABRII), Agricultural Research, Education and Extension Organization (AREEO), Karaj, Alborz, Iran
[4] Faculty of Biological Science, Tarbiat Modares University, Tehran, Tehran, Iran
[5] Department of Microbial Biotechnology and Biosafety, Agricultural Biotechnology Research Institute of Iran (ABRII), Agricultural Research, Education and Extension Organization (AREEO), Karaj, Alborz, Iran
[6] Agricultural Biotechnology, Anand agricultural University, Anand, Gujarat, India

Corresponding author
Mohammad Reza Ghaffari,
Mrghaffari52@gmail.com

## ABSTRACT

Cytokinin oxidase/dehydrogenase (CKX), responsible for irreversible cytokinin degradation, also controls plant growth and development and response to abiotic stress. While the *CKX* gene has been studied in other plants extensively, its function in cotton is still unknown. Therefore, a genome-wide study to identify the *CKX* gene family in the four cotton species was conducted using transcriptomics, quantitative real-time PCR (qRT-PCR) and bioinformatics. As a result, in G. *hirsutum* and G. *barbadense* (the tetraploid cotton species), 87 and 96 *CKX* genes respectively and 62 genes each in G. *arboreum* and G. *raimondii*, were identified. Based on the evolutionary studies, the cotton *CKX* gene family has been divided into five distinct subfamilies. It was observed that *CKX* genes in cotton have conserved sequence logos and gene family expansion was due to segmental duplication or whole genome duplication (WGD). Collinearity and multiple synteny studies showed an expansion of gene families during evolution and purifying selection pressure has been exerted. G. *hirsutum CKX* genes displayed multiple exons/introns, uneven chromosomal distribution, conserved protein motifs, and cis-elements related to growth and stress in their promoter regions. *Cis*-elements related to resistance, physiological metabolism and hormonal regulation were identified within the promoter regions of the *CKX* genes. Expression analysis under different stress conditions (cold, heat, drought and salt) revealed different expression patterns in the different tissues. Through virus-induced gene silencing (VIGS), the *GhCKX34A* gene was found to improve cold resistance by modulating antioxidant-related activity. Since *GhCKX29A* is highly expressed during fibre development, we hypothesize that the increased expression of *GhCKX29A* in fibres has significant effects on fibre elongation. Consequently, these results contribute to our understanding of the involvement of *GhCKXs* in both fibre development and response to abiotic stress.

# INTRODUCTION

Cotton (*Gossypium hirsutum*. L) is an important industrial crop, accounting for about 35% of world's fibre consumption (*Hamid et al., 2019*). However, the production and quality of cotton fibres are affected by harsh environmental conditions such as drought, salinity, extreme temperatures and heavy metal pollution (*Anwar et al., 2022*; *Li et al., 2020*). By regulating physiological and biochemical responses, phytohormones such as cytokinins (CK) play a decisive role in enabling plants to withstand both biotic and abiotic stresses (*Ahmed et al., 2018*; *Wang et al., 2020*; *Xiao, Zhao & Zhang, 2019*). Cytokinins, which are chemically similar adenine derivatives known as N6-substituted adenine derivatives, are particularly important for various plant processes, including cell division, differentiation and morphogenesis (*Hnatuszko-Konka et al., 2021*). These processes encompass a wide range of activities throughout the life cycle of a plant, such as seed germination, leaf senescence, flower development and fruit ripening (*Sharma, Kaur & Gaikwad, 2022*).

Maintaining the delicate balance of cytokinin levels in plants requires a complex interplay of different processes. These processes include cytokinin synthesis enabled by the enzyme isopentenyltransferase (IPT), cytokinin activation, cytokinin inactivation by O-glucosyltransferase, cytokinin reactivation by β-glucosidase and cytokinin degradation catalysed by cytokinin oxidase/dehydrogenase (CKX) enzymes. Among these processes, CKX enzymes play a crucial role in the irreversible degradation of cytokinins (*Thu et al., 2017*). Extensive research on different plant species has highlighted the importance of *CKX* genes in growth, development and response to abiotic stress (*Cáceres et al., 2023*). Studies in *Arabidopsis*, rice, maize and soybean have revealed various functions of *CKX* genes in regulating growth and adaptation to stress (*Le et al., 2012*; *Vyroubalová et al., 2009*; *Werner et al., 2010*; *Yeh et al., 2015*). In maize, *ZmCKX1* plays a crucial role in regulating active cytokinin levels during root development, contributing to stress tolerance (*Zalabák et al., 2014*). In soybean, water deficiency and salt stress induce the expression of *GmCKXs* (*Du et al., 2023*). The upregulation of *CKX* genes under stress conditions leads to an increased degradation of active cytokinins, resulting in a reduction of cytokinin levels in plant cells. This reduction in cytokinin levels allows plants to adapt physiological and biochemical processes to mitigate the effects of stress (*Li et al., 2019a*). For example, overexpression of *CKX2* in tobacco enlarges the root system and improves tolerance to drought stress (*Werner et al., 2010*). Similarly, overexpression of the *CKX* from *Medicago sativa* enhanced salt stress tolerance of *Arabidopsis* (*Li et al., 2019a*). The modulation of endogenous cytokinin levels by stress induced CKXs (SICKXs) also indirectly influences hydrogen production in the tomato leaf (*Cueno et al., 2012*). *Liu et al. (2023)*, used virus-induced gene silencing (VIGS) to show that *GhCKX6b-Dt* induces the antioxidant system and alleviates salt stress in cotton. *Li et al. (2023b)*, also reported that *GhCKX14* in G. *hirsutum* plays an important role in the response to drought stress by modulating the activity of antioxidant enzymes. In addition, *Xu et al. (2019)* discovered that down-regulation of *GhCKX3* delays defoliation and reduces ethylene response. These

results emphasise the important role of *CKX* genes in stress adaptation, root development, antioxidant response and ethylene regulation in different plant species.

In addition, *CKX* genes play a multifaceted role that goes beyond their involvement in defense against environmental stress; they exert multiple effects on plant growth and development. For example, studies in rice have shown that the *OsCKX11* gene affects grain size, leaf senescence and source-sink ratio, highlighting the wide-ranging influence of *CKX* genes on plant physiology (*Zhang et al., 2021*). Similarly, overexpression of *AtCKX1* and *AtCKX3* in transgenic plants resulted in reduced inflorescence size and reduced ability to form floral primordia compared to wild-type plants (*Werner et al., 2003*). In cotton, recent studies have begun to uncover the multiple roles of *CKX* genes in physiological processes such as leaf defoliation, fruit branch internode elongation and stress responses (*Naveed et al., 2023*). Using comprehensive analyses such as RNA-Seq techniques, various studies have shown that *CKX* genes are upregulated in response to treatments with abscisic acid (ABA) and thidiazuron (TDZ), indicating their involvement in cotton leaf defoliation (*Zhou et al., 2022*). Transcriptome profiling has also shown that *CKX* genes are associated with fruit branch elongation in upland cotton (*Ju et al., 2019*). Furthermore, researchers have used RNAi techniques to improve carpel development and ovule formation by downregulating the *GhCKX3b* and *GhCKX3* genes, resulting in higher seed and fibre yields (*Zeng et al., 2022*; *Zhao et al., 2015*). Moreover, several studies indicate a correlation between the development and elongation of cotton fibres and the content of endogenous CKs in the bolls and fibres (*Ahmed et al., 2018*; *Wang et al., 2020*; *Xiao, Zhao & Zhang, 2019*). Genetic manipulation techniques such as overexpression of IPT to increase cytokinin levels or silencing of *GhCKX* transcripts to reduce cytokinin degradation have been used to modulate cytokinin levels in transgenic cotton plants. While overexpression of IPT has no effect on fibre yield and quality, silencing of *GhCKX* transcripts results in increased seed number and slightly increased fibre yield (*Jones & Schreiber, 1997*; *Zhao et al., 2015*; *Zhu et al., 2018*).

*CKX* genes are encoded by a multigene family, and have been phylogenetically and functionally characterized in a variety of plants species, such as such as *Arabidopsis thaliana* (*Schmülling et al., 2003*), *Oryza sativa* (*Zheng et al., 2023*), *Glycine max* L., *Nicotiana tabacum* (*Zheng et al., 2023*), *Medicago truncatula* (*Wang et al., 2021*), *Triticum aestivum* (*Jain et al., 2022*), finger millet (*Eleusine coracana)* (*Blume et al., 2022*), *Liriodendron chinense* (*Sun et al., 2023*), *Brassica oleracea* (*Zhu et al., 2022*) and *Brassica napus* (*Liu et al., 2018*), which underline their indispensable role in plant growth and development. Several studies have indicated a correlation between cotton fibre elongation and cytokinin concentration, wherein cytokinins promote fibre differentiation before flowering but inhibit fibre elongation after flowering (*Chen et al., 1997*). It is hypothesized that CKX enzymes influence fibre elongation by regulating the cellular levels of cytokinins. Given this premise, CKX may exert a significant influence on cotton the promotion of fibre elongation and responses to various stresses. While some *CKX* genes have been identified, the comprehensive understanding of the *CKX* gene family, their phylogenetic relationships, expression patterns during fibre developmental stages, and their involvement in cotton physiology remains elusive. Therefore, this study aims to investigate

the *CKX* gene family in two cultivated allotetraploid cotton species, namely upland cotton (*Gossypium hirsutum*) and sea island cotton (*Gossypium barbadense*), together with their putative genome donors, *Gossypium arboreum* and *Gossypium raimondii*. A comprehensive analysis was carried out using bioinformatics tools to analyse various aspects such as gene structure, chromosomal distribution, spatio-temporal expression patterns, collinearity, *cis*-regulatory elements and gene replication of the *GhCKX* genes. In addition, the expression patterns of *GhCKX* in different tissues and their responses under abiotic stress conditions were analysed. In addition, two genes, *GhCKX29A* and *GhCKX34A*, were selected for further in-depth studies based on their expression profiles in different tissues, developmental stages and responses to cold stress. These findings establish a robust groundwork for future investigations into the involvement of *CKXs* in cotton fibre development and their responses to stress.

## MATERIALS AND METHODS

### Identification and sequence analysis of CKX family genes

We used Cotton FGD (https://cottonfgd.net/) for obtaining the whole-genome sequencing data from four cotton species: *G. barbadense* (version ZJU 2.1), *G. hirsutum* (version ZJU 2.1), *G. arboreum* (version CRI 1.0) and *G. raimondii* (version JGI 2.0) (*Shuya et al., 2023*). HMMER software (http://hmmer.org/) was used to search for predicted CKX proteins in the cotton dataset using the hidden Markov model profiles of the CKX domain (PF01565 and PF09265) (Pfam, https://www.ebi.ac.uk/interpro/entry/pfam/#table). Further the NCBI CDD tool (https://www.ncbi.nlm.nih.gov/Structure/cdd/wrpsb.cgi) was used to check the domain structure of the search results and obtain additional information about the CKX domains found. Next, we compared the *GhCKX* genes obtained from G. *hirsutum* (version ZJU 2.1) with ICR, NAU, JGI, ZM24 version 1.0 and HAU and found no difference. The EXPASY bioinformatics resource portal (https://web.expasy.org/compute_pi/) was employed for estimating the isoelectric points and molecular weights (*Duvaud et al., 2021*). WOLF PSORT (https://www.genscript.com/wolf-psort.html?src=leftbar) was used to determine subcellular localization (*Horton et al., 2007*).

### Phylogenetic analysis

Multiple sequence alignments of the acquired genes were performed using ClustalW and MEGA (MEGA11) to examine the evolutionary relationships among the *CKX* genes (*Tamura, Stecher & Kumar, 2021*). On the basis of the data obtained through comparison and using the neighbour-joining method, the phylogenetic tree was constructed. Four cotton species (G. *arboreum*, G. *barbadense*, G. *hirsutum*, G. *raimondii*), along with *Arabidopsis thaliana*, *Brassica napus*, *Triticum aestivum*, *Oryza sativa*, and *Glycine max*, were included to investigate the evolutionary relationships between *CKX* genes. Additionally, homologous sequences of subtilisins from the four cotton species were obtained using the previously described method. Multiple sequence alignments were performed with ClustalW, and the resulting alignments were utilized to construct evolutionary trees in MEGA11, employing the maximum likelihood method (ML). Finally,

the phylogenetic tree was visualized using ITOL (http://itol.embl.de/) (*Letunic & Bork, 2016*).

## CKX genes' structure and conserved motif analysis

The Gene Structure Display Server (GSDS) (http://gsds.cbi.pku.edu.cn/) was used to determine the gene structure in the cotton genome (G. *hirsutum*) by using genomic sequences  and coding sequences (CDS) as input files. The obtained CKX protein sequences were uploaded to Motif Elicitation (MEME) (https://meme-suite.org/meme/tools/meme), an online tool for motif identification. To gain a deeper understanding of the *CKX* gene family, TBtools was used to envisage the Newick format (NWK) file from the phylogenetic study, gene structure maps from MEME, conserved protein motifs and the *G. hirsutum* GFF3 genome file (*Chen et al., 2020a*).

## Analyses for studying location on chromosome, gene evolution and cis-acting elements

The full genome annotation data for *Gossypium barbadense* and *hirsutum* (ZJU), *G. arboreum* (CRI) and *G. raimondii* (JGI) were downloaded from the Cotton FGD. To find the orthologous gene pairing between the sea-island or upland cotton genomes and the diploid ancestors of A/D cotton species, Blast version 2.2.9 was used (*Huo et al., 2023*). Genomic collinearity blocks were analyzed using the MCScanX software program. TBtools software was used to determine the chromosomal map positions and the *CKX* gene replication in cotton (all four species) (*Li et al., 2023a*). The non-synonymous (Ka) and synonymous (Ks) substitutions rates for the duplicated genes was estimated using TBtools software to study the evolutionary selection pressure on the *CKX* genes (*Zhao et al., 2022*).

For *cis*-element analysis to explore gene expression regulation, the 2 kb region upstream of the start codon for all *CKX* genes was employed as the promoter sequence. PlantCARE (Cis-Acting Regulatory Element) (https://bioinformatics.psb.ugent.be/webtools/plantcare/html/) was utilized to examine the *cis*-elements within the promoter regions of *GhCKX* genes, and the obtained data were processed using TBtools software.

## Evaluation of *CKX* gene expression in different tissues and under different stress conditions

Transcriptome data from the Cotton Omics Database (http://cotton.zju.edu.cn/) were used to investigate the expression profiles of *CKX* genes under different abiotic stress conditions and tissue types. In addition, previously unpublished RNA-seq data of fibre samples from five different developmental stages and from two genotypes with different fibre qualities were used. The creation of heat maps was facilitated by TBtools.

## Plant materials used, treatments provided

The seeds of the non-genetically modified varieties TABLA and Tab11 of *G. hirsutum* underwent a pre-germination phase in sand at 25 °C for 4 days. The seedlings were then transplanted into a hydroponic system with the Hoagland nutrient solution (*Hothem, Marley & Larson, 2003*). In the greenhouse, the daytime temperature was set to

28 °C and the nighttime temperature to 25 °C. The photoperiod lasted 16 h and the relative humidity fluctuated between 60% and 70%. Once the cotton seedlings reached the trifoliate stage, they were subjected to cold stress by transferring them to an environment with a temperature of 4 °C under normal light conditions. Leaf samples were harvested at 0, 3, 6, 12 and 24 h intervals after stress for RNA extraction. Each treatment was replicated three times. The collected leaf samples were quickly frozen in liquid nitrogen and stored at −80 °C until RNA extraction

## RNA extraction and qRT-PCR analysis

The hot borate RNA isolation protocol, as described by *Hamid et al. (2020)*, was employed to extract total RNA from frozen leaves. The extracted RNA was then converted to cDNA using the Quantitect reverse transcriptase kit (Qiagen, Hilden, Germany). For qPCR analysis, the Taq Pro Universal SYBR qPCR Master Mix (Qiagen, Hilden, Germany) was utilized in conjunction with an ABI 7500 Fast Real-Time PCR instrument (Thermo Fisher Scientific, Waltham, MA, USA). Based on the power analysis (performed to ensure statistical robustness), 30 samples were chosen for real time qPCR analysis. Three biological replicates were included for each experimental condition. The $2^{-\Delta\Delta Ct}$ method, with *GhUBQ7* as the internal reference gene (*Li et al., 2023c*), was employed to analyze the qPCR data. Statistical analyses were conducted using SPSS 22.0 (SPSS, Chicago, IL., USA), and the relative expression level data of the target gene were subjected to one-way analysis of variance (ANOVA) using two-tailed tests. The results were presented as means ± standard error (SE). Statistical significance was determined at the levels of *$P < 0.05$, **$P < 0.01$, and ***$P < 0.001$. A list of the primer sequences can be found in Table S1A.

## Virus-induced gene silencing and cold stress

The silenced *GhCKX34A* fragment was designed using the SGN VIGS tool (https://vigs.solgenomics.net/) and subsequently inserted into the pTRV2 vector using specific primers designed with https://www.ncbi.nlm.nih.gov/tools/primer-blast/. The primers used for amplification are listed in Table S1B. We used the TM-1 cDNA library for upland cotton to amplify the specific segment of *GhCKX34A*. We then transformed the constructs into *Agrobacterium tumefaciens* strain LBA4404. Following the protocol described by *Mustafa et al. (2016)*, we performed virus-induced gene silencing (VIGS) and maintained cotton seedlings under specific growth conditions. Both VIGS and control plants were treated at 4 °C for 24 h. We then collected the leaves for biochemical analyses. Malondialdehyde (MDA) and hydrogen peroxide ($H_2O_2$) content were quantified using the MDA (EEA015) and $H_2O_2$ (23280) assay kits (Thermo Fisher Scientific). Each experiment was performed in triplicate.

# RESULTS

## Identification of members of the *CKX* gene family in cotton

Through this work, we detected 307 *CKX* genes in four species of *Gossypium*: 62 genes each in the diploid species (G. *arboreum* and G. *raimondii*), 87 in G. *hirsutum*, and 96 in G. *barbadense*. Both diploid species has the same number of *CKX* genes, but the tetraploid

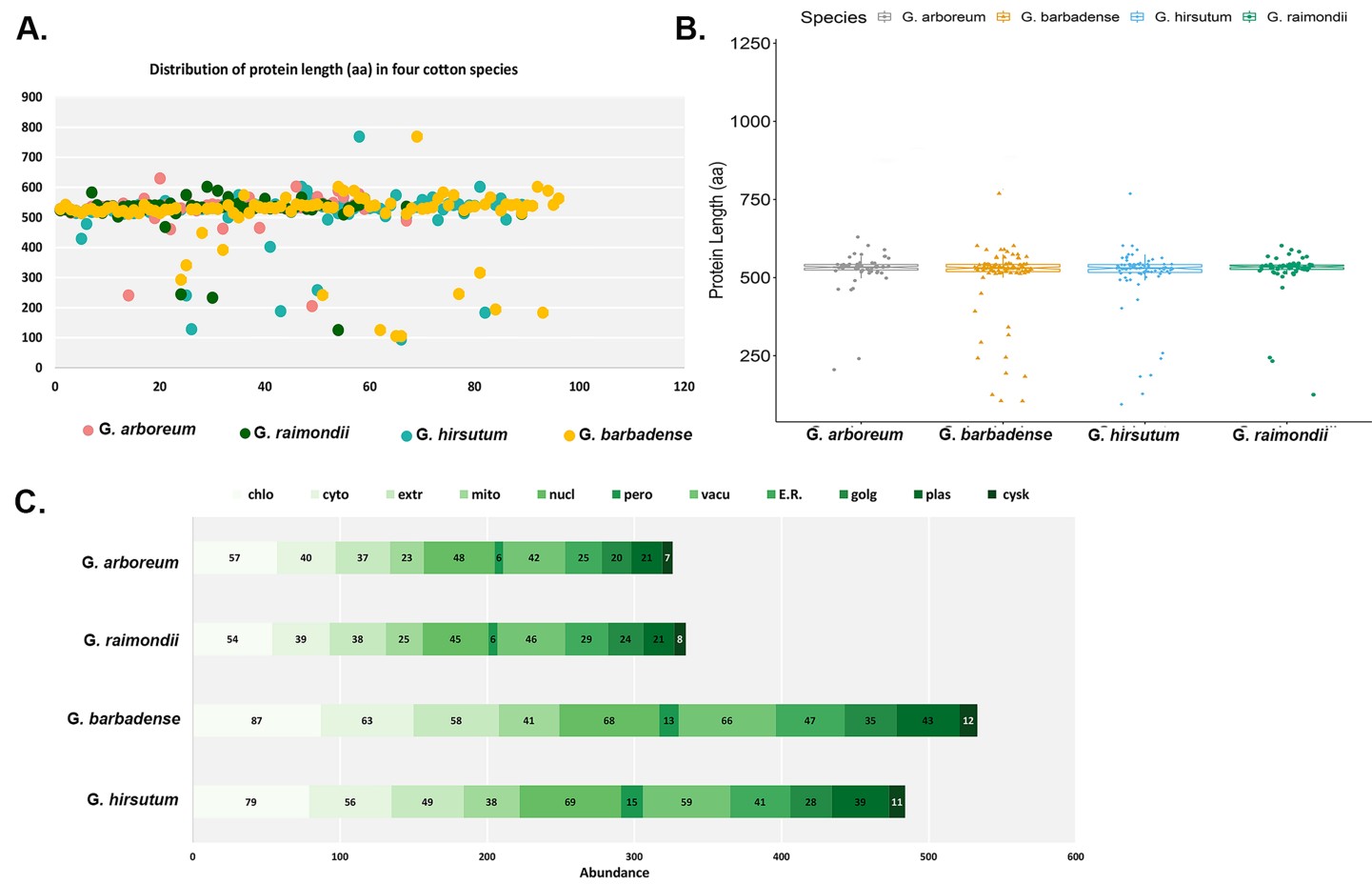

**Figure 1 Properties of CKX proteins in four cotton species.** (A) Length distribution, (B) molecular weight distribution and (C) subcellular localisation of CKX proteins in four cotton species.

species of cotton (G. *hirsutum* and *barbadense*) have less than twice as many as the diploid species (G. *raimondii* and *arboreum*). Compared to G. *barbadense*, G. *hirsutum* had the fewer *CKX* genes, showing the influence of polyploidisation and hybridisation in allotetraploid cotton species. We further renamed all the *CKX* gene family members we obtained, based on their chromosomal location (Table S2, and File S1).

The main features of the identified *CKX* genes were then predicted and are shown in Fig. 1. The results show that 62 *CKX* genes of G. *arboreum* encode peptides whose length ranged from 205 to 630 amino acids, their average being 524 aa (Fig. 1A). They occupy 0.0823% of the genome. The molecular weights of *GaCKX* peptides spanned across a range of 23.15 to 70.03 kDa (Fig. 1B). The isoelectric points ranged from 5.353 to 9.942. Most *GaCKX* genes were located in the chloroplast, with only a few in the peroxisome, plasma membrane, or Golgi (Fig. 1C). Protein lengths of G. *raimondii CKX* genes varied from 125 to 602 amino acids, with an average length of 519 aa and a total length of 32,239 aa and an occupied position in the genome of 0.0961% (Fig. 1A). The peptides molecular weights ranged from 14,121 to 68,489 kDa (Fig. 1B). These proteins had isoelectric points ranging from 5.102 to 10.032, with an average of 7.67, indicating that they are slightly alkaline.

The majority of *GrCKX* genes were found to be situated in the chloroplast and nucleus, while a limited number were identified in the peroxisome or mitochondria (Fig. 1C).

The protein lengths of the *CKX* genes of the allotetraploid cotton G. *hirsutum* ranged from 94 to 769 amino acids (AA), with their average being 509 AA, a total length of 44,314 AA and an occupancy rate of 0.0431% in the genome (Fig. 1A). The molecular weights and the isoelectric points of *GhCKX* peptides began from 10,193 to 85,359 kDa (Fig. 1B) and 5,212 to 9,897, respectively. Most *GhCKX* genes were detected in chloroplasts, with the exception of four genes that were detected in peroxisomes (Fig. 1C). The protein lengths of the *CKX* genes of G. *barbadense* were from 105 to 769 AA, with a mean length of 504 AA, an average length of 828 AA and a total length of 19,968 AA (Fig. 1A). 0.0513% of these genes were located in the genome. The peptides of *GbCKX* have molecular weights between 11,388 and 85,344 kDa and isoelectric points between 4,417 and 9,897 (Fig. 1B). Subcellular localization studies showed that most genes were identified in the chloroplast, followed by the nucleus, and that the few genes were detected in the cytoskeleton and peroxisome (Fig. 1C).

## Phylogenetic analysis of *CKXs*

To investigate the cotton *CKX* genes' evolutionary relationship, Clustal W in MEGA11 software was utilized for the comparison of 87 CKX protein sequences of G. *hirsutum*, and a neighbour-joining approach was employed to make a rootless phylogenetic tree. In this phylogeny, cotton *CKX* genes were randomly divided into five distinct subfamilies, CKX I-V (Fig. 2). The pink represents CKXI, the largest subfamily with 47 *CKX* genes. CKXII-CKXV consists of four, 27, three, and six *CKX* genes, respectively. Genes within the same subgroup are presumed to exhibit similar or identical functions. The location of most CKX proteins of homologous chromosomal subgroups A and D was observed on the same branch.

To understand the phylogenetic relationship between the *CKX* genes of the four cotton species with *Arabidopsis thaliana*, *Brassica napus*, *Triticum aestivum*, *Oryza sativa*, and *Glycine max*, we constructed a phylogenetic tree containing 463 protein sequences from G. *arboreum* (62), G. *raimondii* (62), G. *barbadense* (96), G. *hirsutum* (87), A. *thaliana* (seven), B. *napus* (36), T. *aestivum* (41), O. *sativa* (17), and G. *max* (55). The phylogenetic tree was randomly divided into nine subclades (Fig. 3). The CKX proteins of these species can be detected in almost all branches. Subclade CKXI contains the most members (225) and subclade CKXV the fewest (27); subclades CKXII, CKXIII, and CKXIV contain 77, 89, and 28 genes, respectively (Fig. 3). Remarkably, homologous proteins were seen in every subclade for almost all the CKX proteins in *Arabidopsis* and the four cotton species, indicating the functional conservation of CKX proteins in dicotyledons. Furthermore, it has already been proven that island and upland cotton have evolved from crosses between G. *arboreum* and G. *raimondii*. This is supported by the observation that CKX proteins are preferentially clustered in both diploid (G. *arboreum* and G. *raimondii*) and tetraploid (G. *hirsutum* and G. *barbadense*) cotton. Moreover, *GbCKX* and *GhCKX* pairs consistently cluster together, highlighting the gene duplication function during evolution.
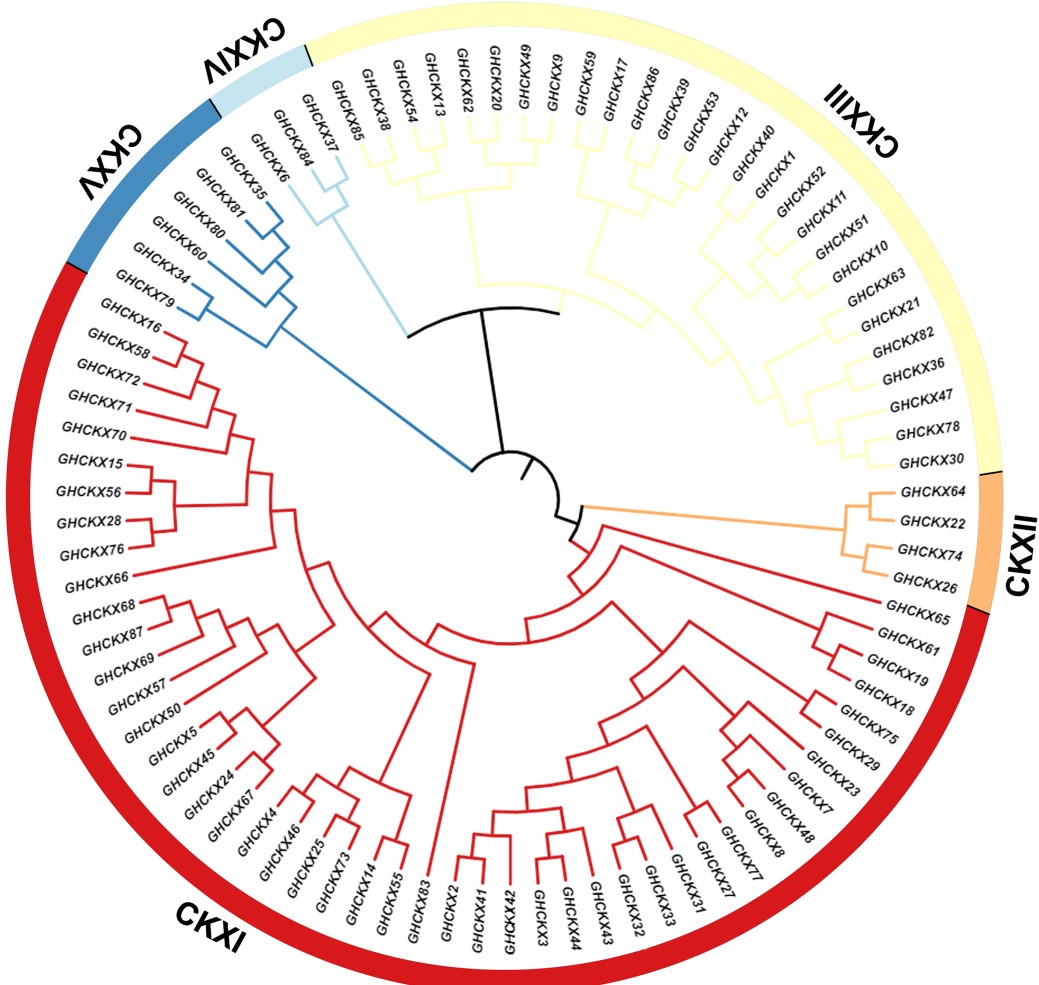

**Figure 2 Two unrooted phylogenetic trees of CKX genes were constructed using MEGA11.** The GhCKX family evolutionary tree was constructed using the neighbour-joining method, and the inter-specific evolutionary tree of CKX genes was constructed using the maximum likelihood method. Phylogenetic tree of CKX family protein sequences in upland cotton.

## Analysis of *CKX* genes duplication, multiple synteny and collinearity

Many plants have experienced polyploidization, an ancient mechanism involving genome-wide duplication that leads to the multiplication of all genes within a specific region of the genome (*Wang et al., 2019*). To investigate the evolution and consequences of polyploidisation and hybridisation, we examined the forms of *CKX* gene duplication in all four cotton species. The *CKX* genes of G. *barbadense*, G. *arboreum*, G. *raimondii* and G. *hirsutum* exhibit either whole genome duplication (WGD) or segmental duplication events. In G. *hirsutum*, there are also dispersed gene duplications in three of the *CKX* genes, tandem duplications in 44 *CKX* genes and proximal duplications in 23 *CKX* genes. In addition, we identified two *CKX* genes from G. *barbadense*, one *CKX* gene from G. *raimondii* and two from G. *arboretum* that displayed a singleton gene duplication type (Table S3).

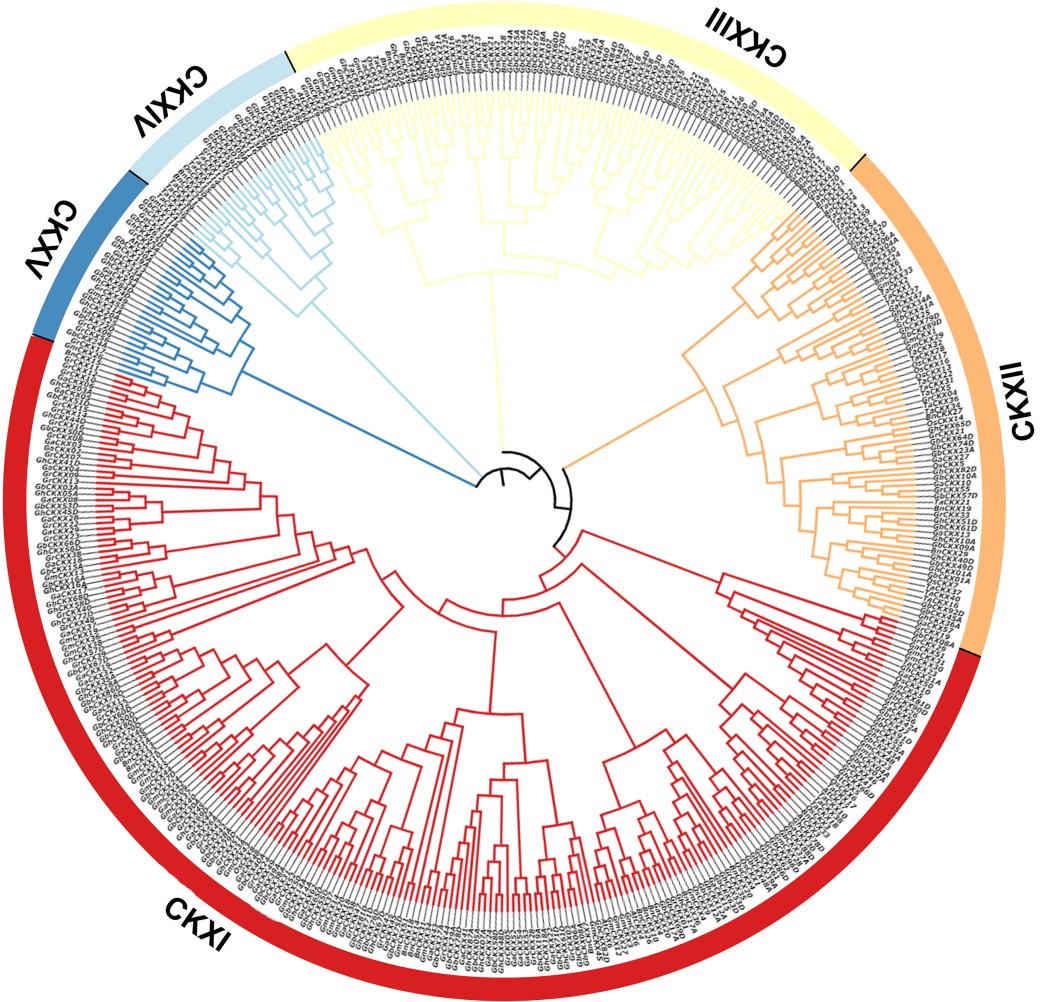

**Figure 3 Phylogenetic relationships of 463 CKX proteins from G. *hirsutum*, G. *barbadense*, G. *arboreum*, G. *raimondii*, *Arabidopsis thaliana*, *Brassica napus*, *Triticum aestivum*, *Oryza sativa*, and *Glycine max*.**

According to a multiple synteny analysis of the *CKX* genes of G. *raimondii*, G. *barbadense*, G. *arboreum* and G. *hirsutum*, there are 67 orthologous gene pairs between G. *arboreum* and *hirsutum*, 75 between G. *raimondii* and *hirsutum*, and between G. *hirsutum* and G. *barbadense* there were 74 orthologous gene pairs (Fig. 4, Table S4). To ascertain the nature of selective pressure acting on these orthologous gene pairs during evolution, the ratios of non-synonymous to synonymous substitutions (Ka/Ks ratios) were computed. The Ka/Ks value was <1 for all orthologous gene pairs, with the exception of one gene pair between G. *hirsutum* and G. *arboreum*, and another gene pair between G. *hirsutum* and G. *raimondii*. Conversely, in homologous gene pairs, all the comparisons resulted in Ka/Ks <1 (Table S4).

To investigate the relationships between locus of subgenome A and subgenome D of G. *barbadense* and G. *hirsutum*, a collinearity analysis was performed (Fig. 5). The Ka/Ks value was less than 1 for 38 orthologous/paralogous pairs in G. *hirsutum* (Fig. 5A;
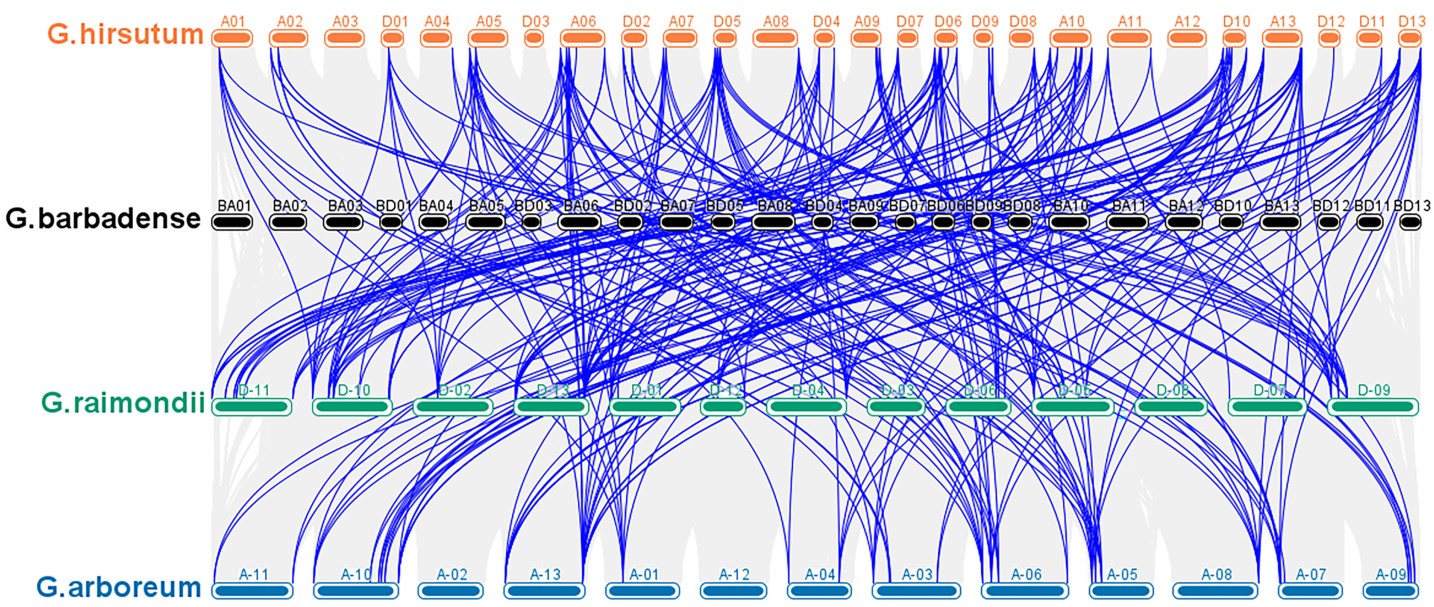

**Figure 4  Multiple synteny analysis among cotton CKX genes.** Multiple synteny analysis was used to show the orthologous relationship among G. *hirsutum*, G. *barbadense*, G. *arboreum*, and G. *raimondii*. Chromosomes of different cotton species were represented with different colors.

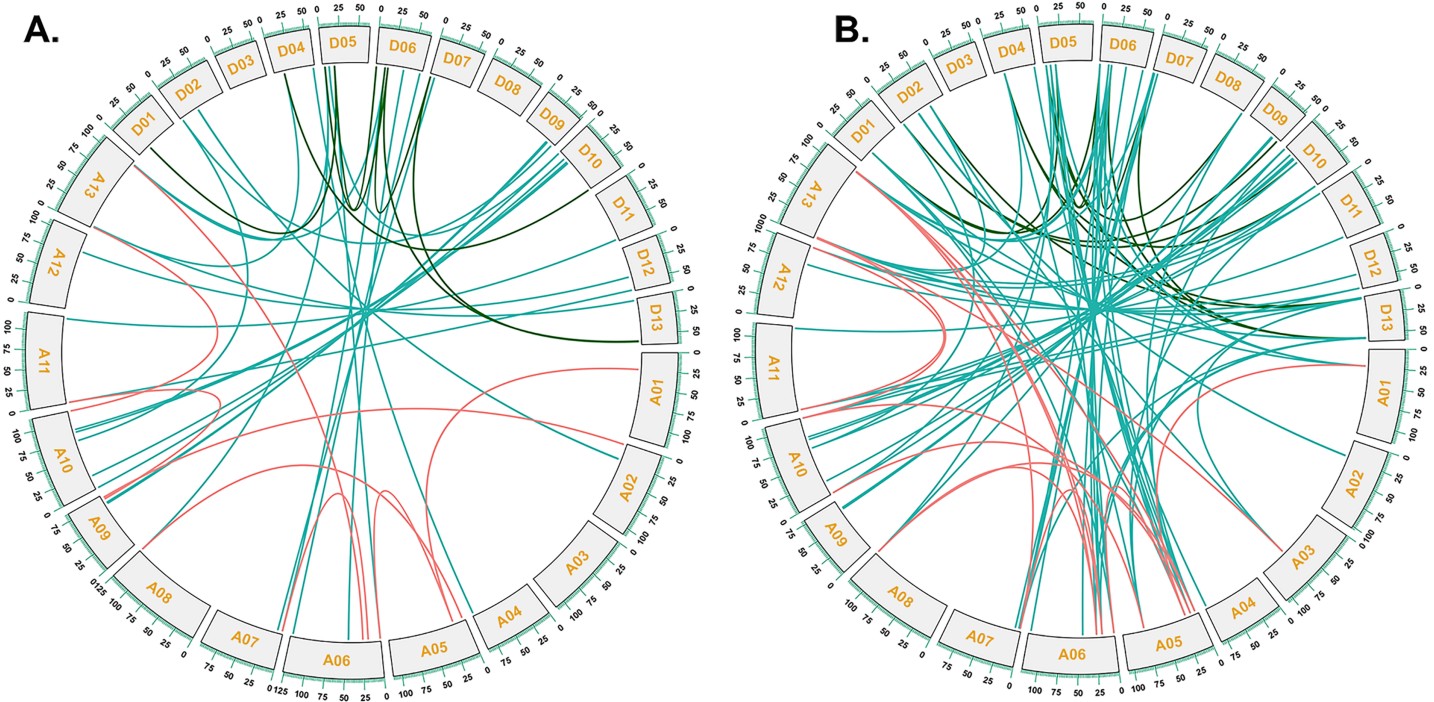

**Figure 5  Collinearity analysis of G. *hirsutum* and G. *barbadense* CKX genes.** (A) Collinearity analysis of G.*hirsutum* CKX genes. (B) Collinearity analysis of *G. barbadense* CKX genes. A01 to A13 represents A-subgenome chromosomes while D01 to D13 represents D-subgenome chromosomes. Homologous gene pairs between A- to A-subgenome were represented with pink lines, homologous gene pairs between A- to D-subgenome were represented with blue lines, and homologous gene pairs between the D- to D-subgenome were represented with green lines.

Table S5). Likewise, in G. *barbadense*, a total of 38 orthologous/paralogous gene pairs were found (Fig. 5B; Table S6) and except one, all the other ortholog/paralog genes had Ka/Ks less than 1 (*GB_D10G0977.1/GB_A10G1006.1*). More specifically, all but two pairs of *GhCKX* genes (*GH_A10G1692.1/GH_D10G1208.1* and *GH_A04G1592.1/ GH_D04G1941.1*) had Ka/Ks values lower than 0.5, whereas Ka/Ks values of 33 *GbCKX* genes was less than 0.5 while for five genes were greater than 0.5. This suggests that there was a high level of purifying selection during the *CKX* gene evolution.

### *GhCKX* gene motif and structure composition correlation analysis

To gain a comprehensive understanding of the potential structural evolutionary relationships among *GhCKX* family members, we scrutinized the exon/intron patterns, phylogenetic trees, structural characteristics, and protein motifs of *GhCKX* genes. A maximum likelihood phylogenetic tree among *GhCKX* genes was clustered in compliance with the exon/intron structure and motif distribution pattern (Fig. 6A). Ten motifs were found in the GhCKX proteins. Based on the pattern of motif distribution, GhCKX proteins with comparable patterns of motif distribution were grouped together (Fig. 6B), indicating that the pattern is largely conserved and that they may perform the same activities. The distribution patterns of coding (CD) sequences, intron, and untranslated regions (UTR) were then determined by gene structure analysis. The architecture of *CKX* genes could be divided into two categories: genes having less introns and those having more introns. The CKXV subfamily members generally contain one to two exons, while others have more exons, ranging from four to 18. More than half of the *GhCKX* genes studied were found to have multiple introns. As shown in Fig. 6C, intron-exon configurations within the same subfamily are comparable.

### Chromosomal location of *CKXs* in the four species of *Gossypium*

The chromosomal locations of *CKX* genes were established in order to more accurately examine distribution of genes on chromosomes as well as gene replication in four *Gossypium* species and the 307 genes were observed to be distributed randomly on all the chromosomes of the four *Gossypium* species. There was a random distribution of the 87 genes on 25 chromosomes in G. *hirsutum*, with one gene situated on the scaffold. Between one and thirteen *CKX* genes were located on each chromosome. There was tandem replication on A04, A07 and A11, and on D02, D07, and D10. Subgenome A had 39 genes and subgenome D had 47 genes. While chromosome D03 did not have any *CKX* gene, indicating the possibility that these putative *CKX* genes were duplicated or eliminated during evolution (Fig. 7A). Similar to G. *hirsutum*, there was a random distribution of 96 genes on 25 chromosomes in G. *barbadense* and also chromosome D03 lacked *CKX* genes, indicating gene duplication. Each chromosome contained between one and fourteen *CKX* genes. Subgenomes A and D each contained 48 genes. The chromosomes A04, A07, A09, A10, A11 and A12 and chromosomes D10 showed tandem replication (Fig. 7B).

In G. *arboreum*, a total of 62 genes were dispersed irregularly across 12 chromosomes, with an additional gene located on a scaffold. Notably, there was at least one *CKX* gene on chromosome A01, and up to 17 genes were identified on chromosome A10. Tandem

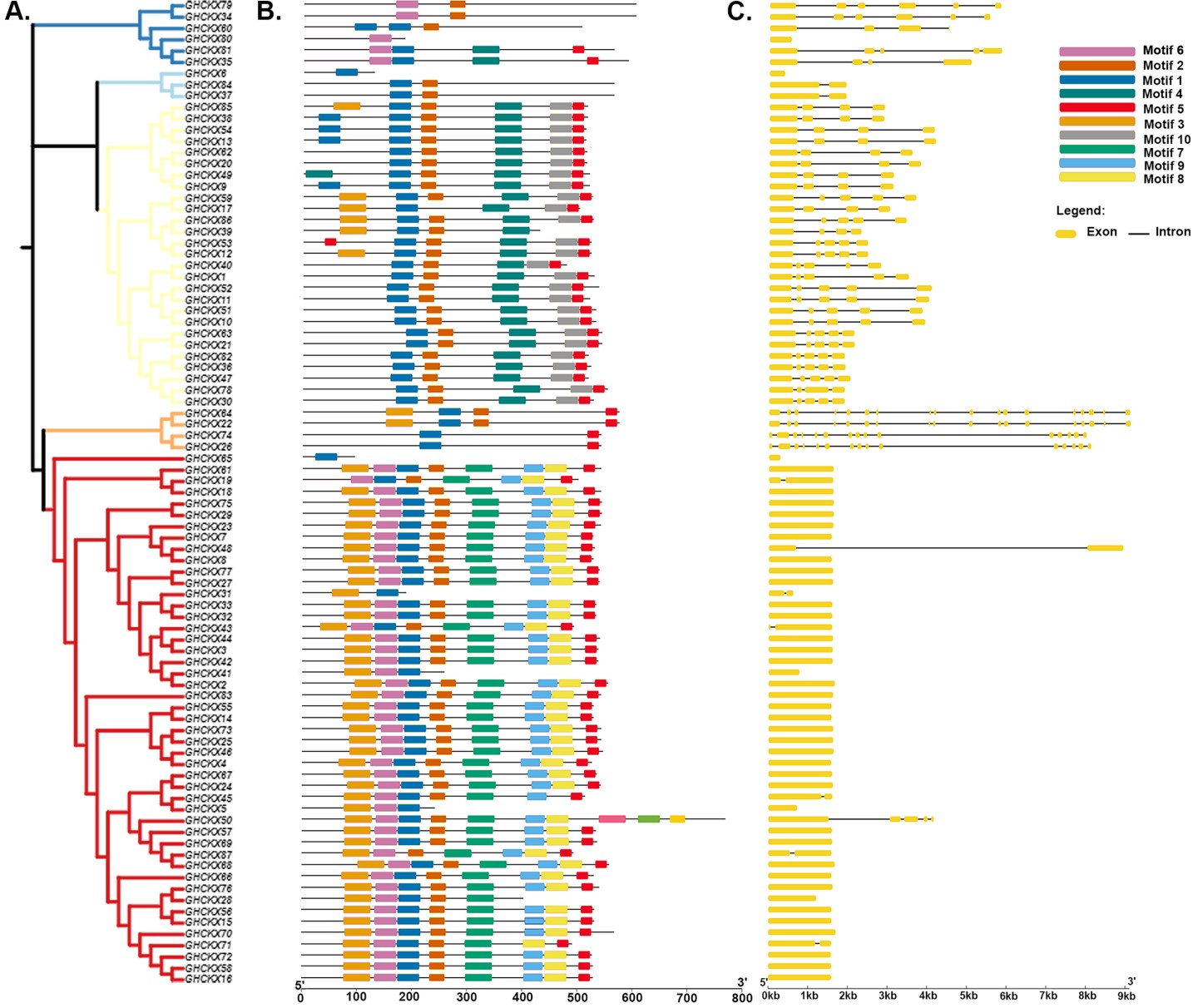

**Figure 6 Genetic structure of the *CKX* gene family in upland cotton.** (A) Phylogenetic tree of the *CKX* gene family. (B) Motif pattern diagram of the *CKX* gene family. (C) Exon structure diagram of the *CKX* gene family.

replication events were observed on Chromosomes 03, 04, 07, 10, 11, and 12, as well as on one scaffold identified as tig00015851 (Fig. 7C). *G. raimondii* has 62 genes distributed on 12 chromosomes, with all genes irregularly distributed. Most *GrCKX* genes were located on chromosomes Chr11 and Chr05, followed by seven *GrCKX* on chromosome Chr10. The chromosome with the fewest *GrCKX* genes was Chr02. No *GrCKX* genes were detected on chromosome Chr03. The chromosomes 04, 05, 07, 11, and 12 showed tandem replication (Fig. 7D). Finally, the predominant gene amplification strategies during the evolution of *CKX* genes were dispersed and fragmentary duplication.

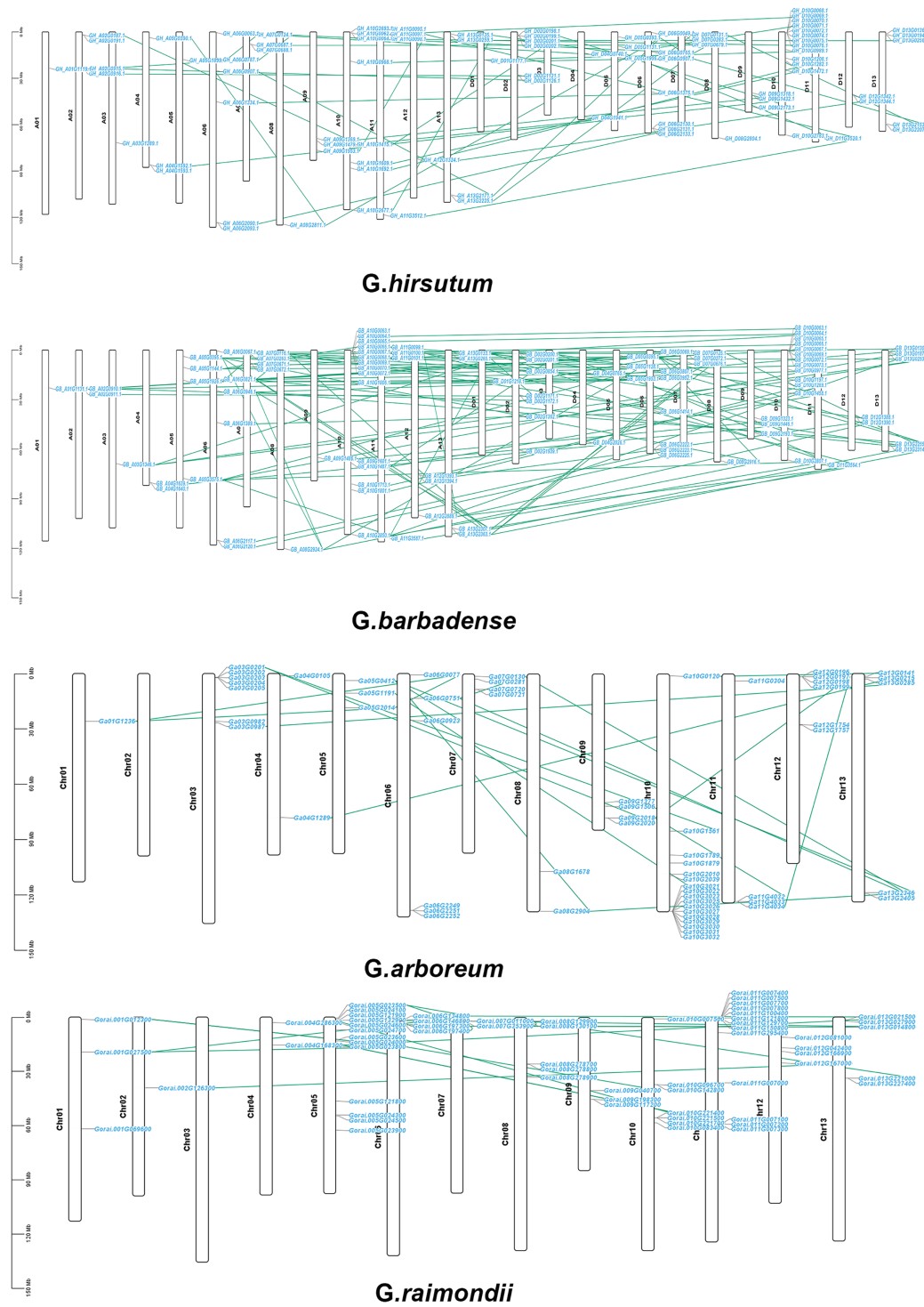

**Figure 7 Chromosomal localization and gene duplication of *CKX* genes in *G. arboretum*, *G. raimondii*, *G. hirsutum*, and *G. barbadense*, and tandem duplication of gene pairs during evolution is shown by lines.**
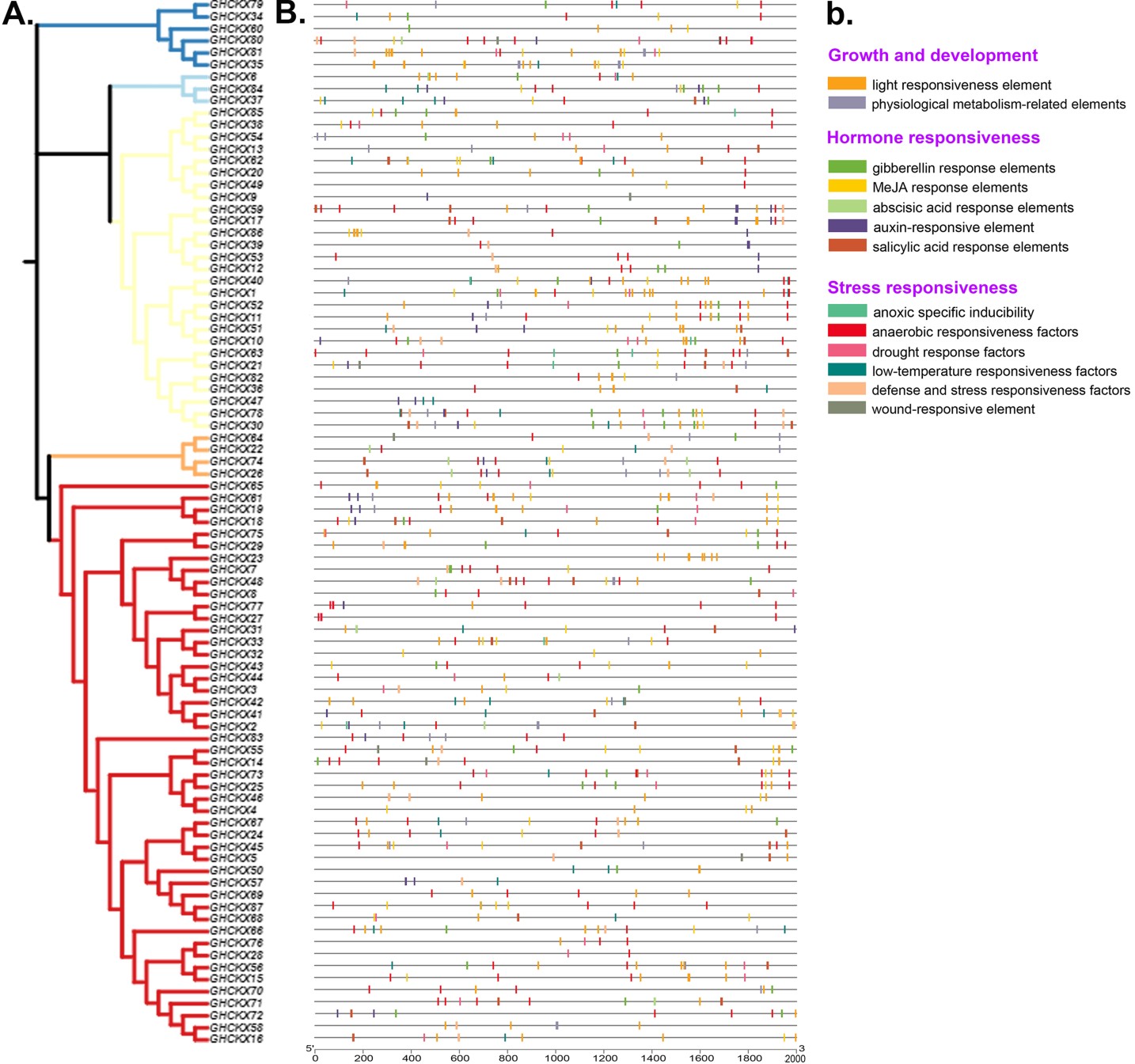

**Figure 8 Analysis of promoters and differentially expressed genes of the *GhCKX* family.** (A) Phylogenetic tree of *GhCKX* genes. (B) Cis-acting elements in the promoters of *GhCKX* genes. (C) The organizational expression of *GhCKX* genes.

## *GhCKX* promoter analysis

Cis-regulatory elements are transcriptional control elements that are involved in various biological processes and stress responses. To better understand gene regulatory processes, we estimated the 2 kb upstream region of *GhCKX* genes using the PlantCARE web tool. For each gene, 15-20 cis-regulatory elements involved in physiological metabolism,

hormones and stress responses were discovered and listed. The most common cis-elements associated with hormones are abscisic acid responsive elements, auxin and salicylic acid responsive elements, gibberellin responsive elements and methyl jasmonate (MeJA) responsive elements. Abiotic stress-responsive elements include stress- and defence-responsive elements, light- and cold-responsive elements, wound-responsive elements and drought-inducing elements. The CAT-Box and MYB DNA binding site (MBSI) have also been discovered in physiological metabolic components. Although unrelated to their subfamilies, practically all promoters contain a large number of hormone response elements (Figs. 8A, 8B). Salicylic acid-regulatory elements, abscisic acid-responsive elements, MeJA-regulatory elements and gibberellin-responsive components are found in the majority of *GhCKXs* promoters. Abscisic acid-responsive elements were found in 64 genes, MeJA-responsive elements in 57, gibberellin-responsive elements in 51, salicylic acid-responsive elements in 33 and auxin-responsive elements in 24 belonging to the *GhCKX* family. In addition, we discovered 34 genes with elements that respond to defence and stress, 33 genes with elements that respond to low temperatures, 25 genes with elements that respond to drought, 23 genes with anoxic inducible elements and 22 genes with elements that respond to wounding. By performing promoter analysis, we could pool genes responsive to various plant hormones and response processes under varying conditions, which enables us to corroborate future gene activities.

## *GhCKXs* expression pattern in various tissues and under abiotic stresses

To examine the expression specificity of *GhCKXs* across different tissues, we employed publicly available transcriptome data (FPKM values) for various tissues, including root, petal, pistil, calycle, leaf, stamen, stem, and torus, to generate a heatmap (Fig. 9A). Most *GhCKX* have shown differential expression among different tissues and many genes were tissue specific. *GHCKX80D*, *GHCKX04A*, *GHCKX03A*, and *GHCKX27A* were strongly expressed in roots, while *GHCKX82D*, and *GHCKX30A* were expressed only in leaves (Fig. 9A).

Previous researches have shown that *GhCKXs* respond to abiotic stressors (*Liu et al., 2023*). In our investigation of *GhCKXs* response, we used publicly available RNA-seq data from TM-1 treated at low and high temperatures and with polyethylene glycol (PEG) and NaCl to examine the expression patterns of *GhCKX* (Fig. 9B). Remarkably, many detected *GhCKX* genes were induced under different abiotic stress factors and showed different expression patterns. The expression of *GhCKX54D* showed significant down-regulation in response to cold, heat, NaCl and PEG stressors. It is noteworthy that the expression patterns of certain *GhCKX* genes slightly coincide within the same group, as observed for *GhCKX64D* and *GhCKX22A* as well as *GhCKX74D* and *GhCKX26A*. To validate the results of the transcriptome analysis, the cotton seedlings were subjected to cold treatment. Subsequently, qRT-PCR was performed for selected genes, including *GHCKX16A*, *GHCKX58D*, *GHCKX34A*, and *GHCKX84D* (Fig. 9C). The findings indicated an upregulation in the expression of *GhCKX16A* and *GHCKX34A* within the first 24 h under low-temperature conditions, reaching a peak after 24 h. These results suggest alterations in

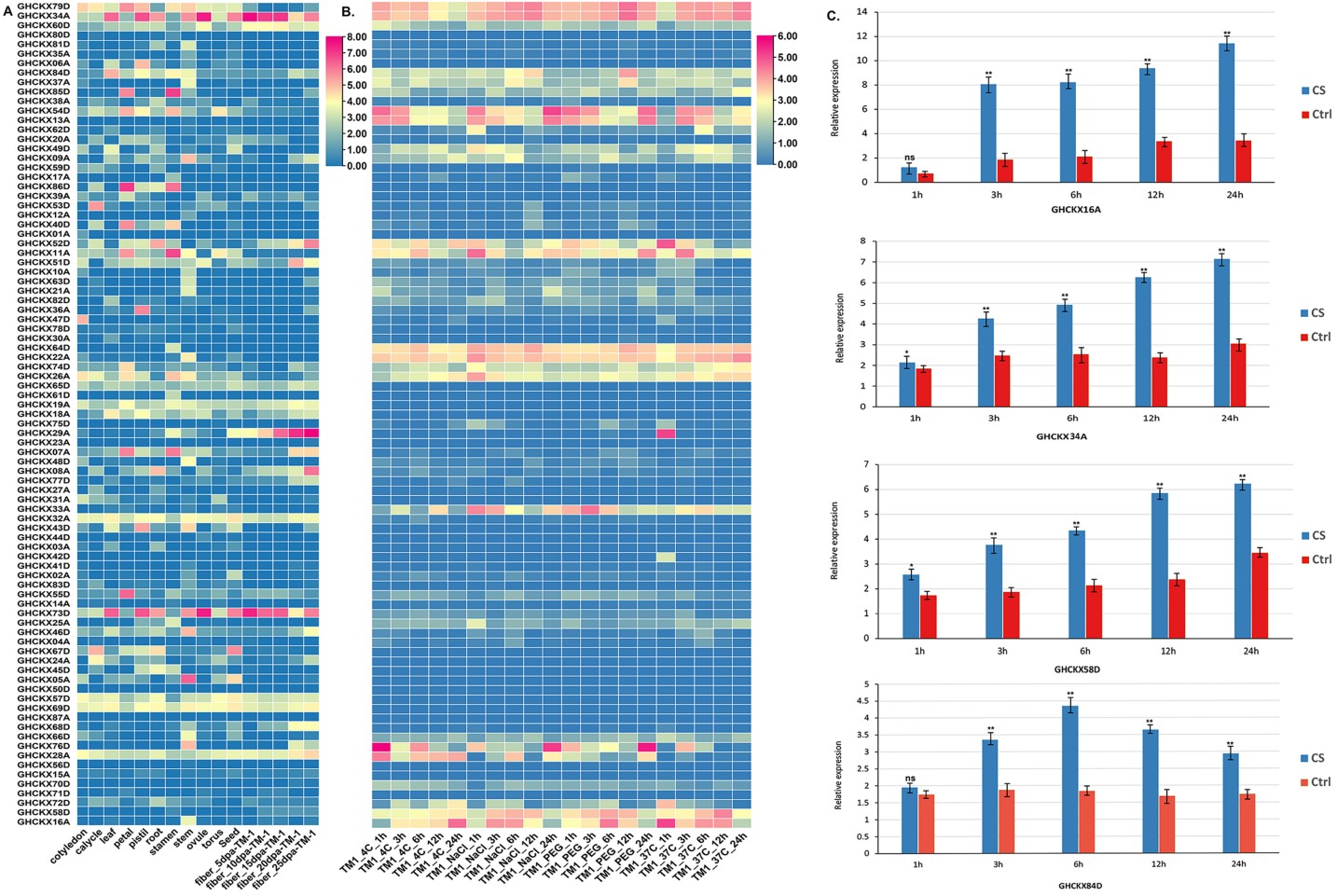

**Figure 9 The expression patterns of GhCKX genes.** (A) Heatmap displaying expression of expressed GhCKX under temperature, hot, PEG and salt stresses. (B) Heatmap displaying expression of expressed GhCKX in each tissue. (C) Relative expression levels of GhCKX genes in cotton under cold stress (CS) and control conditions (Ctrl) in leaf tissues. *$p < 0.05$, **$p < 0.01$, and ns (not statistically significant) (Student's one-tailed t test).

the expression patterns of several *GhCKXs* following the treatment, highlighting their potential role in enhancing adaptability to chilling stress (Fig. 9C).

## Gene ontology analysis of the GhCKXs

To improve our understanding of the functions of *GhCKXs*, we performed functional enrichment in Gene Ontology (GO) using the AgriGo v2 website (http://systemsbiology. cau.edu.cn/agriGOv2/) and a cut-off value of ≤0.01 for the *p*-value. This approach provided a more detailed insight into gene functions and included a large number of significantly enriched terms. The results of the GO-BP enrichment analysis showed eight terms such as cellular proteins, metabolic process (GO:0044267), regulation of response to stress (GO:0080134), and organization of single-organism membranes (GO:0044802). GO-CC enrichment revealed three terms such as ribosome (GO:0005840), part of the Golgi apparatus (GO:0044431) and apoplast (GO:0048046). GO-MF enrichment uncovered 17 terms including ion binding (GO:0043167), nucleotide binding (GO:

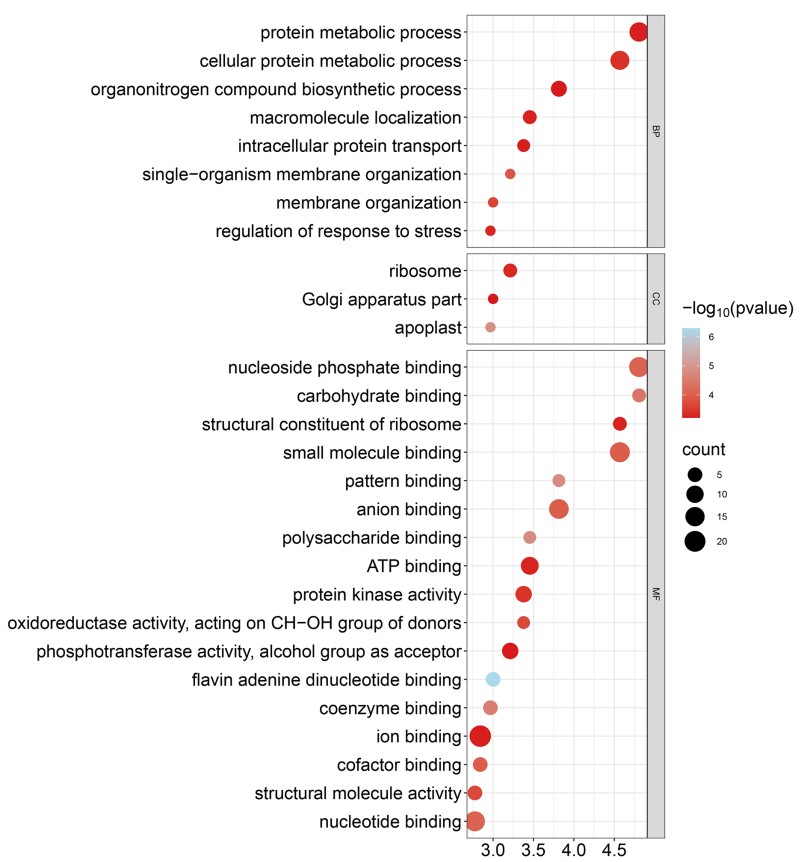

**Figure 10 Bubble plot showing GO enrichment analysis of *GhCKXs*.** The top 20 GO terms significantly enriched by GhCKXs.               

0000166), nucleoside phosphate binding (GO:1901265) and anion binding (GO:0043168) (Fig. 10, Table S7). In summary, the outcomes of the Gene Ontology (GO) enrichment analysis validated the involvement of *GhCKXs* in various biological processes related to the regulation of stress response, anion binding, and membrane components.

## *GhCKX*s role in fibre development

In assessing the impact of *GhCKXs* on cotton fibre development, our study focused on investigating the expression dynamics of *GhCKXs* at different fibre development stages in two different samples, namely TABLA and Tab11, each differing by unique fibre lengths and thicknesses. The expression levels of *GhCKX29A* and *GhCKX34A* were particularly conspicuous in the fibre tissues of TABLA, Tab11 and TM-1, suggesting that *GhCKX29A and GhCKX34A* occupies central role in the complicated process of cotton fibre development (Fig. 11A). To deepen our understanding of the involvement of *GhCKX29A* and *GhCKX34A* in fibre development, we closely examined the fluctuations of *GhCKX29A* and *GhCKX34A* expression in the two samples by qRT-PCR (Fig. 11B). Our results showed a progressive increase in the expression of *GhCKX29A* and *GhCKX34A* from 5 DPA to 25 DPA in both samples, which is in seamless agreement with the transcriptome data previously obtained for TM-1. Of note, the expression of *GhCKX29A* and *GhCKX34A* in

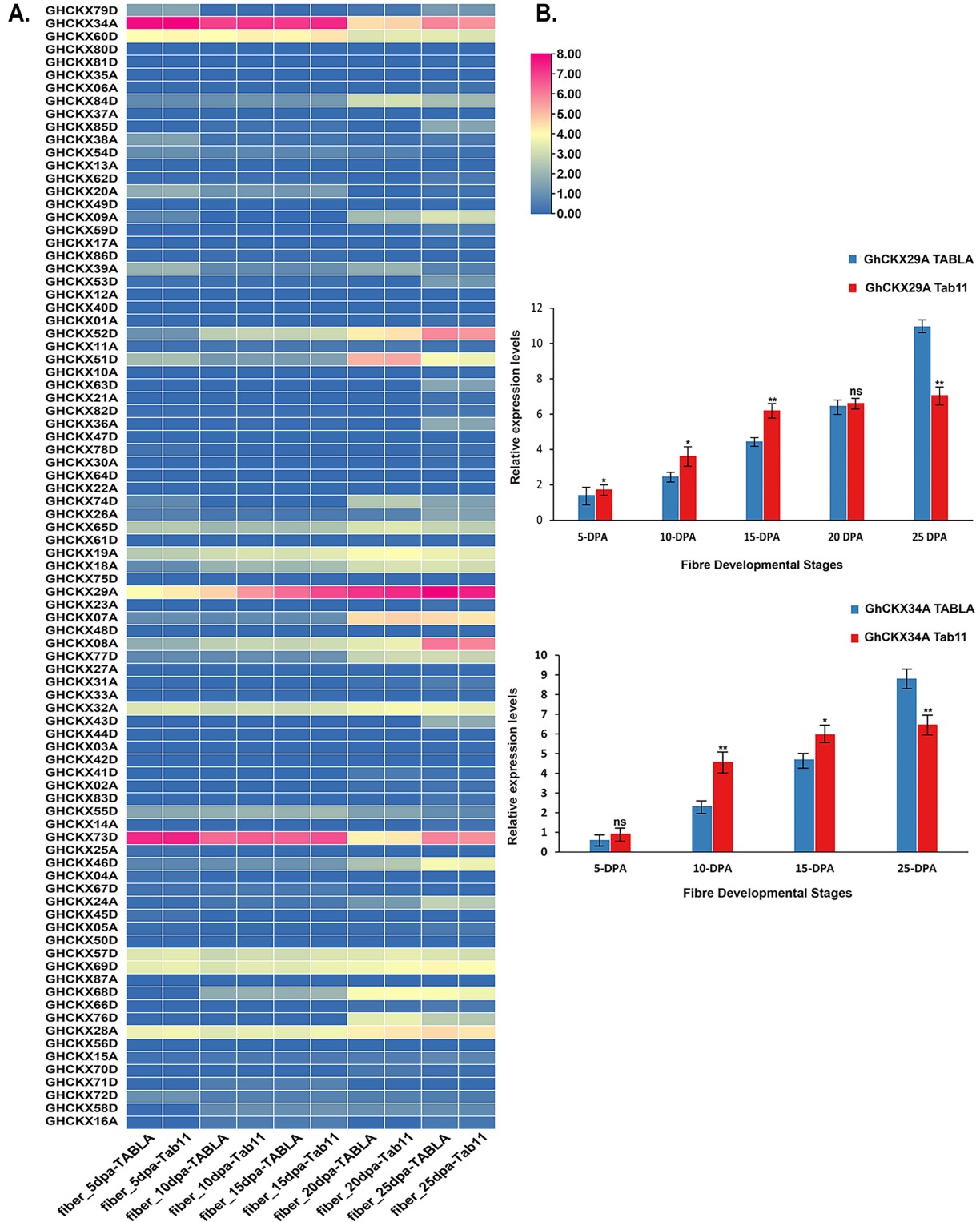

**Figure 11 Expression patterns of *GhCKXs* in cotton fiber.** (A) The expression of GhCKXs of TABLA and Tab11 at different fiber developmental stages. (B) qRT-PCR results of GhCKX29A and GhCKX34A at different fiber developmental stages. *$p < 0.05$, **$p < 0.01$, and ns (not statistically significant) (Student's one-tailed $t$ test).

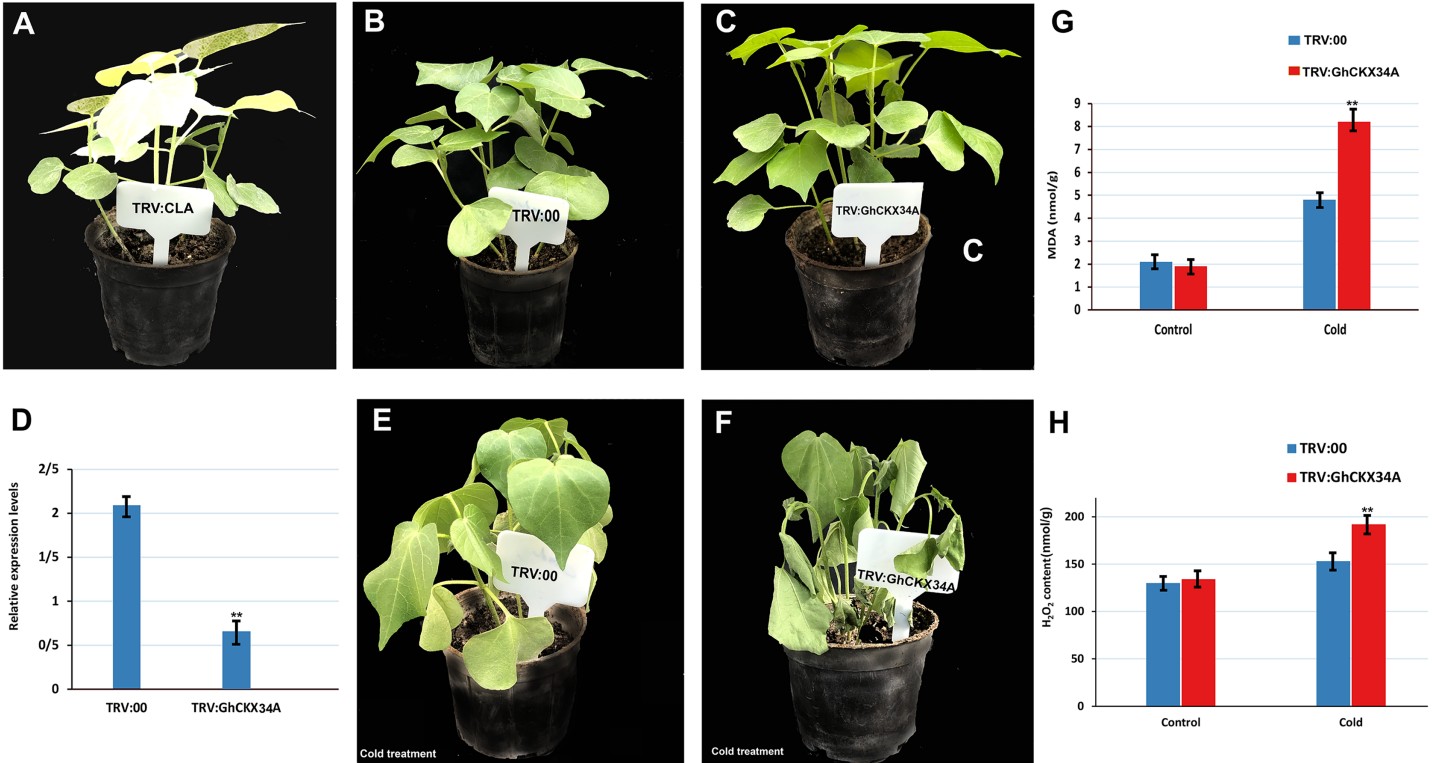

**Figure 12  Silencing *GhCKX34A via* virus-induced gene silencing (VIGS) enhances the cold resistance regulation of cotton.** (A) Plant albino phenotypes of TRV::CLA. (B, C) Phenotypes of TRV::00 and TRV::*GhCKX34A* before cold stress. (D) *GhCKX34A* expression levels in leaves of TRV::00 and TRV::*GhCKX34A* plants. (E, F) Phenotypes of TRV::00 and TRV::*GhCKX34A* after cold treatment for 24 h. (G, H) The MDA and H2O2 content of TRV::00 and TRV::*GhCKX34A* after cold stress. *$p < 0.05$, **$p < 0.01$, and ***$p < 0.001$ (Student's one-tailed t test).

Tab11 was higher than in TABLA at 5 DPA, 10 DPA and 15 DPA, while it lagged significantly behind TABLA at 25DPA. These findings strongly indicate the potential involvement of these genes in influencing the elongation of cotton fibres.

## Silencing *GhCKX34A* decrease tolerance to cold stress

To investigate the role of *GhCKX34A* in the response to cold stress, we performed a VIGS assay to reduce *GhCKX34A* expression in TABLA plants. The albino phenotype ensured the success of tobacco rattle virus (TRV)::CLA1 in cotton (Fig. 12A), and comparison of the expression level of TRV::00 and TRV::*GhCKX34A* in cotton showed that gene expression had been successfully suppressed (Figs. 12B–12D). Subsequently, plants infiltrated with TRV::*GhCKX34A* and TRV::00 constructs were subjected to cold stress for 24 h. No visible differences in appearance were observed between the control plants (Fig. 12b) and the plants infiltrated with TRV::*GhCKX34A* (Fig. 12C) before the cold treatment. After 24 h of cold treatment, a significant phenotypic difference in leaf damage was observed between TRV::00 (Fig. 12E) and TRV::*GhCKX34A* (Fig. 12F) plants. The MDA and H2O2 content in TRV::*GhCKX34A* plants was about 1.7 and 1.3 times higher, respectively, than in TRV::00 plants (Figs. 11G, 11H). These results indicate that silencing of the *GhCKX34A* gene exacerbates the susceptibility of cotton to cold.

## DISCUSSION

CTK is associated with a variety of plant physiological processes, like leaf senescence, seed fatty acid production (*Wu et al., 2017*), flower organ development and pod formation (*Liu et al., 2016*), and seed yield (*Murai, 2014*). The hormone also contributes to plant responses to a number of abiotic stresses, like salinity (*Joshi et al., 2018*), temperature (*Bielach, Hrtyan & Tognetti, 2017*), and drought (*Golan et al., 2016*). CKXs encoded by a small gene family have been studied in numerous plant species, including finger millet (*Eleusine coracana*) (*Blume et al., 2022*), rice (*Zheng et al., 2023*), wheat (*Triticum aestivum* L.), Chinese cabbage (*Liu et al., 2013*), maize (*Zea mays* L.), canola (*Brassica napus* L.), and alfalfa (*Medicago sativa* L.). However, there are only limited reports on *CKX* genes in cotton (G. *hirsutum*). In this investigation, we conducted a detailed study of CKX genes across four cotton species. To identify evolutionary relationships, phylogenetic investigations, protein motifs, sequence logo analysis, localization on chromosome, gene structure, multiple synteny, gene duplication and collinearity research were carried out. The role of *GhCKX* genes was determined using tissue-specific expression analysis, cis-element analysis and response to cold stress.

### Evolution of *CKX* genes in cotton

In each of the species G. *arboreum* and G. *raimondii*, 62 *CKX* genes were detected, respectively, in G. *hirsutum* 87 *CKX* genes, while G. *barbadense*, 96 *CKX* genes were found. The variations in the *CKX* gene number between species may be due to genome evolution and replication, resulting in homologous genes synthesis and a rise in their number (*Liu et al., 2021*). In comparison with other plant species, the *CKX* gene family in G. *barbadense*, G. *arboreum*, G. *raimondii* and G. *hirsutum* is the largest, with 30 in *Oryza sativa* (*Zheng et al., 2023*), 36 in *Arabidopsis* (*Schmülling et al., 2003*), 28 in *Sorghum bicolor* (*Mameaux et al., 2012*), 18 in *Glycine max* L (*Nguyen et al., 2021*), and 11 in foxtail millet (*Setaria italica*) (*Wang, Liu & Xin, 2014*). There is a correlation between the genome sizes of the four species, which are 885 Mb for G. *raimondii* (*Udall et al., 2019*), 1,746 Mb for G. *arboreum* (*Huang et al., 2020*), 2,173 Mb for G. *hirsutum*, and 2,224.98 MB for G. *barbadense* (*Meng et al., 2023*) and the ratio of *CKX* members. In addition, there was an inconsistency in the total number of *GhCKX* genes compared to the combined gene numbers of G. *raimondii* and G. *arboretum*. This could be associated with the complicated recombination and transposition of the Dt and At subgenomes, which could lead to the loss and inactivation of many genes (*Pei et al., 2022*). These genes were branched five in the phylogenetic tree, each containing *CKX* genes from cotton and *Arabidopsis*.

The comparison between genomes is a quick and effective method to investigate the probable functions and properties of genes (*Bayer et al., 2020*). Therefore, we can infer the putative role of homologous *CKX* genes in cotton by examining the *CKX* genes data in *Arabidopsis*, the model plant. The common positioning of five orthologous gene pairs in subclades of the phylogenetic tree between cotton and *Arabidopsis* genomes can confirm this, implying that CKX proteins are functionally conserved in these dicotyledonous plants. For instance, the *atCKX1* gene expression, associated with the lateral roots formation, is observed in root tissues (*Del Bianco, Giustini & Sabatini, 2013*).

The simultaneous localization of this gene with *CKX* genes of cotton species may be indicative of the putative functions of *CKX* genes in cotton, but their functions have to confirmed further through reverse genetics approaches in future. Detailed molecular characterization of *CKX* genes revealed a great diversity among the members of this gene family. The physicochemical features of CKX protein members, such as molecular weight (MW), protein length, and isoelectric point (pI), align with findings from previous studies conducted in canola (*Liu et al., 2018*), and maize (*Brugiere et al., 2003*). These variations underscore the functional diversity within this gene family.

The analysis of *GhCKX* gene structures show that they have multiple introns and exons, and the *CKX* members of a subgroup, that are closely related, have similar gene structure and domain composition, which may indicate comparable developmental functions in the plant. Previous studies have found that the number of exons/introns of *CKX* gene family members ranges from 1 to 4 in wheat (*Jain et al., 2022*), and from 1 to 10 in soybean (*Du et al., 2023*). In this study, the number of exons/introns in G. *hirsutum* ranged from 8 to 18, which is consistent with the number of exons/introns in *Phaseolus vulgaris* (*Zhang et al., 2023*), suggesting that the exons/introns of *CKX* genes were deleted or inserted during the evolution of G. *hirsutum*. The structural differences between exons and introns are the consequence of insertions or deletions and are crucial for studying how the gene family evolved (*Li et al., 2019b*). Numerous genome-wide studies indicate that the process of intron gain or loss was widespread during the diversification of eukaryotes (*Gao et al., 2022*). Differences in the length of introns between genes showed that they have important functions in the *GhCKX* genes' functional divergence. In addition, there were 10 conserved protein motifs with slight variations in protein motifs in *GhCKX* genes that might be linked to growth and abiotic stress tolerance in plant. Analysis of the protein motifs revealed that some motifs are specific to a particular group and provide information about the functions of that group. It was also found that motif 1 is common to all CKX proteins of G. *raimondii*, G. *arboreum*, G. *hirsutum* and G. *barbadense*.

Studies related to gene distribution on chromosomes revealed that *GaCKX* and *GrCKX* genes in G. *raimondii* and G. *arboreum* are irregularly dispersed on 12 chromosomes. *GbCKX* was found on 13 At and 12 Dt subgenome chromosomes in G. *barbadense*. Genes duplicated on the same chromosome (two or more) confirmed a tandem duplication event, while those with duplication on different chromosomes were referred to as segmental duplications (*Gerdol, Greco & Pallavicini, 2019*). Examination of the chromosomal location of *GhCKX* genes showed that *GhCKX* genes were unevenly distributed among various chromosomes and this uneven type of distribution on A *and* D subgenomes maybe due to deletion or addition of genes due to WGD or segmental duplications and also incomplete genome sequencing. Some chromosomes such as D01, D08, D11 and A01, A03, A08 and A12, have only one gene. Most of the genes (thirteen *GhCKX* genes) were located on chromosome D10. In addition, the *GhCKX* genes consisted of several *cis*-elements in their promoter region associated with circadian control, light response, auxin response, abscisic acid response, phytochrome regulatory elements, zein metabolism, low temperature, anaerobic induction, meristem and endosperm expression, elements responding to gibberellin, MeJA, and salicylic acid. Earlier studies identified the
light-induced cis-elements GT1 motif, G-box, I-box and AT-rich regions (*Sun et al., 2023*), AuxRE, DR5, the cis-elements induced by auxin (*Li et al., 2022*), and the cis-elements CATGTG and CACG induced by cold (*Bhadouriya et al., 2021*). The abundance of elements in the promoter region of *GhCKX* genes suggests the functional diversity of these genes in cotton. Furthermore, *GhCKX* genes harbor *cis*-elements in their promoter regions associated with a stress response (low-temperature and drought response) and hormone response elements (salicylic acid response, auxin response, abscisic acid response, gibberellin response, and methyl jasmonate (MeJA) response elements) were consistent with prior studies in soybean (*Du et al., 2023*) and wheat (*Jain et al., 2022*). The occurrence of different elements in the *GhCKX* genes' promoter region indicate their functional diversity in cotton

### *CKX* gene duplication and expansion

Gene duplications are important for the diversity of the plant genome, as they lead to the emergence of new genes and genetic regulatory pathways. During evolution, gene duplication may have aided plants in adapting to different abiotic stresses (*Panchy, Lehti-Shiu & Shiu, 2016*). The most important cause of gene family expansion is gene duplication (tandem as well as segmental gene duplication) (*Kamburova et al., 2021*). Several former studies have shown that tandem and segmental duplications have a major role in the expansion of several plant gene families (*Zhang et al., 2020*; *Zhao et al., 2020*). Distribution and duplication analysis showed that the *CKX* genes of G. *arboreum* and G. *raimondii* had WGD or tandem duplications, but we also noticed two *CKX* genes having singleton gene duplication. Remarkably, our gene duplication analysis in alloploid cotton species reveal that segmental duplication was probably the major reason for gene expansion in G. *hirsutum* and G. *barbadense* (74.8% and 78.5%, respectively). Most Ka/Ks ratios were below 1.0 which suggested that cotton *CKX* gene family was subjected to high purifying selection pressure and had undergone limited functional divergence. A subsequent examination of the locus relationships between the G. *hirsutum* A subgenome of and the G. *barbadense* D subgenome revealed 10 paralogous/orthologous *GhCKX* gene pairs in G. *hirsutum* with Ka/Ks value lower than 1 while in G. *barbadense*, there were 28 paralogous/orthologous genes with Ka/Ks less than 1. The Ka/Ks ratio gives information on the selection pressure to which the duplicated genes were subjected over time. Ka/Ks = 1.0 means duplicated gene pairs have undergone neutral selection, Ka/Ks > 1.0 points to a positive selection during rapid evolution and Ka/Ks < 1.0 indicates purifying selection. In G. *hirsutum*, due to the occurrence of purifying selection, the proliferation of *GhCKX* family genes was suppressed, which increased fixation, lowered the extent of loss-of-function mutations that were deleterious at duplicated loci, and also preserved the role of newly duplicated genes (*Wu et al., 2023*).

### Expression profile analysis of *GhCKX* genes

The level of gene expression in the tissues and organs is directly linked to their functions (*Xia et al., 2022*). The gene expression analysis of *GmCKXs* in soybean by qRT-PCR revealed expression patterns that were specific to particular tissues and developmental

stages (*Du et al., 2023*). To assess the expression profiles of the 87 *GhCKX* genes in diverse cotton tissues, we utilized both publicly available datasets data. *GhCKX* gene expression altered between cotton tissues, implying that *GhCKX* genes have different biological roles and contribute to the regulation of cotton growth and various tissue development. *GhCKX34A* and *GhCKX60D* had strong expression in all organs, *GhCKX03A* was identified to be expressed mainly in the root, and the expression of *GhCKX72D* decreased at the stages of fibre maturation, consistent with the study by *Zeng et al. (2022)*. Previous studies have shown that *BnCKX5-1*, 5-2, 7-1 and 7-3 have higher expression in oilseed rape leaves, while *BnCKX1-2*, 1-3 and 1-4 are more highly expressed in flowers (*Liu et al., 2018*). However, most of *GhCKX* genes are moderately to weakly expressed in leaves but strongly expressed in roots. These findings suggest a potential correlation between *GhCKX* genes and the root development of cotton plants.

Cis-acting elements are required for transduction of signal and initiation of gene transcription. Exploration of the cis-acting regions of the *GhCKX* promoter has shown that *GhCKXs* are involved in growth and development of plants, hormone response and response to biotic and abiotic stresses. Comparable results were observed for *CKX* gene families in soybean, canola, millet, Arabidopsis, and maize (*Li et al., 2022*). For example, exogenous application of hormones in bread wheat, had a significant effect on the *TuCKXs* gene expression within a few hours (*Shoaib et al., 2019*). Ectopic expression of the alfalfa (*Medicago sativa*) *MsCKX* gene in *Arabidopsis* improved the salt tolerance of the transgenic plants, whereas exogenous treatment with 6-BA inhibited the expression of *GmCKX* in soybean (*Du et al., 2023*). Transcriptome analysis of *GhCKX* genes under different stresses suggests that specific *GhCKX* genes exhibit distinct expression patterns in response to cold, drought, and salinity stress. Further validation using qRT-PCR showed that, at the seedling stage and under low-temperature stress, *GHCKX16A* and *GHCKX34A* were up-regulated in the leaves after 24 h. *CKX*-overexpressing *Arabidopsis* lines exhibit increased frost tolerance compared to wild-type plants, suggesting that these receptors act as inhibitors of low-temperature stress (*Tiwari et al., 2023*). The silencing of *GHCKX34A* indicated that the VIGS plants were more sensitive to cold, as measured by MDA and $H_2O_2$ content (Fig. 12). The MDA content in TRV:: *GHCKX34A* was significantly higher than that of TRV::00 after 24 h cold treatment, indicating that the plasma membrane damage was more severe in the TRV:: *GHCKX34A* plant (Fig. 12). These results coincided with the previous research (*Li et al., 2019a*; *Liu et al., 2023*). The transcriptomic results suggested that *GHCKX34A* is also involved in the regulation of cotton tissue, ovule and fibre growth and development. Furthermore, we hypothesised that *GHCKX34A* members are associated with abiotic resistance in cotton, particularly cold stress resistance. However, the underlying molecular mechanism needs to be further elucidated.

Cytokinin exerts influence on diverse facets of plant development, impacting processes such as cell division, senescence in plant tissues and organs, and the regulation of apical dominance; however, several studies suggest that it suppresses both growth and elongation of fibres (*Zeng et al., 2019*). *GhCKX29A and GhCKX34A* were found to be strongly expressed in fibres and differentially expressed in TABLA and Tab11 fibres at each developmental stage (5, 10, 15, 20, and 25 DPA). Studies have shown that inhibition of

*CKX* expression, which plays a negative regulatory role, can increase endogenous cytokinin levels in plants (*Chen et al., 2020b*). TABLA is a chromosome segment substitution line (CSSL) with different genetic backgrounds, created by crosses between upland cotton Tab11 as a recurrent parent and Sea Island cotton 92001. It is characterised by an exceptional fibre quality achieved through successive backcrosses of high generations and selection. The fibre strength and length of TABLA is better than that of Tab11. In Tab11, the level of expression of these genes during fibre development were higher in 5 DPA, 10 DPA and 15 DPA than in TABLA. However, at 25 DPA, the expression of these genes were much higher in TABLA. It has been suggested that the level of endogenous cytokinin and the first three fibre development stages are lower in Tab11 than in TABLA, which may be necessary for fibre development (*Xiao, Zhao & Zhang, 2019*). Later, the cytokinin content was lower in TABLA than in Tab11, which enabled the accumulation of auxin in the ovary epidermis and promoted fibre elongation. Studies have indicated that high concentrations of kinetin (>5 µM), a cytokinin type, impede fiber elongation, while lower concentrations (<0.5 µM) promote it (*Beasley & Ting, 1974*; *Yu et al., 2000*). Furthermore, fiber elongation was suppressed in transgenic cotton expressing the cytokinin biosynthesis isopentenyltransferase gene, *IPT*, controlled by the seed-specific promoter Ph/P (*Yu et al., 2000*). another investigation revealed that constitutive overexpression of GhCKX-RNAi had minimal adverse effects on fiber quality, including length, strength, and fineness (*Zhao et al., 2015*). Therefore, *GhCKX29A* and *GhCKX34A* may affect fibre quality by modulating endogenous cytokinins. (*Zeng et al., 2022*).

## CONCLUSION

In this study, 307 *CKX* genes were identified in four cotton species, 87 of them in G. *hirsutum*. Phylogenetic analysis categorised these genes into five distinct groups and revealed that the expansion of the *CKX* gene family in cotton was significantly influenced by either segmental or whole-genome duplication events. The duplicated *CKX* genes showed conserved amino acid sequences, suggesting a purifying selection pressure during evolution. Analysis of the *GhCKX* gene structure revealed conservation with multiple protein motifs and exons/introns. Interestingly, the distribution of *GhCKX* genes in subgenomes A and D was irregular. Furthermore, our study highlighted the central role of *GhCKX* genes in regulating cotton growth, with their expression patterns being associated with flower, root and fibre development. Notably, qRT-PCR validation under various abiotic stress treatments emphasized the involvement of *GhCKX* genes in stress responses, particularly under cold stress conditions. A focused investigation into the expression of *GhCKX29A* and *GhCKX34A* in fibres of varying lengths and strengths revealed its significant contribution to the process of fibre elongation. This study particularly highlights the significance of the *GhCKX34A* gene in enhancing cotton cold tolerance by modulating the antioxidant enzyme activities. In addition, genomic insights into *CKX* genes offer the potential for marker-assisted selection (MAS) and genomic selection (GS), facilitating the rapid integration of favourable *CKX* gene variants into elite cotton germplasm. This has promising implications for the development of high-yielding, stress-tolerant cotton varieties with superior fibre quality traits. These results provide a

solid foundation for further research into *CKX* genes in cotton stress response and fibre development.

### Funding

This study was supported by the Cotton Research Institute of Iran (CRII) and funded under the project (0138-07-0705-017-0004-02040-020782) of the Agricultural Biotechnology Research Institute of Iran (ABRII). The funders had no role in study design, data collection and analysis, decision to publish, or preparation of the manuscript.

### Grant Disclosures

The following grant information was disclosed by the authors:
Cotton Research Institute of Iran (CRII): 0138-07-0705-017-0004-02040-020782.
Agricultural Biotechnology Research Institute of Iran (ABRII).

### Competing Interests

Sushil Kumar is an Academic Editor for PeerJ.

### Author Contributions

- Rasmieh Hamid conceived and designed the experiments, performed the experiments, analyzed the data, prepared figures and/or tables, authored or reviewed drafts of the article, and approved the final draft.
- Feba Jacob performed the experiments, analyzed the data, authored or reviewed drafts of the article, and approved the final draft.
- Zahra Ghorbanzadeh performed the experiments, analyzed the data, prepared figures and/or tables, authored or reviewed drafts of the article, and approved the final draft.
- Mojtaba Khayam Nekouei conceived and designed the experiments, authored or reviewed drafts of the article, and approved the final draft.
- Mehrshad Zeinalabedini conceived and designed the experiments, authored or reviewed drafts of the article, and approved the final draft.
- Mohsen Mardi conceived and designed the experiments, authored or reviewed drafts of the article, and approved the final draft.
- Akram Sadeghi conceived and designed the experiments, authored or reviewed drafts of the article, and approved the final draft.
- Sushil Kumar conceived and designed the experiments, authored or reviewed drafts of the article, and approved the final draft.
- Mohammad Reza Ghaffari conceived and designed the experiments, analyzed the data, authored or reviewed drafts of the article, and approved the final draft.

## Data Availability

The data is available in the Supplemental Files.

The bioinformatics tools and resources used in this study are available at:

- HMMER software: (http://hmmer.org/)
- Pfam (https://www.ebi.ac.uk/interpro/entry/pfam/#table)
- NCBI CDD tool (https://www.ncbi.nlm.nih.gov/Structure/cdd/wrpsb.cgi)
- EXPASY bioinformatics resource portal (https://web.expasy.org/compute_pi/)
- WOLF PSORT (https://www.genscript.com/wolf-psort.html?src=leftbar)
- ITOL (http://itol.embl.de/)
- Gene Structure Display Server (GSDS) (http://gsds.cbi.pku.edu.cn/)
- Motif Elicitation (MEME) (https://meme-suite.org/meme/tools/meme)
- PlantCARE (Cis-Acting Regulatory Element) (https://bioinformatics.psb.ugent.be/webtools/plantcare/html/)
- Cotton Omics Database (http://cotton.zju.edu.cn/).

## Supplemental Information

Supplemental information for this article can be found online at http://dx.doi.org/10.7717/peerj.17462#supplemental-information.

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
