# Peer review of "Genomic insights into CKX genes: key players in cotton fibre development and abiotic stress responses"

_PeerJ, doi:10.7717/peerj.17462_

## Round 0.1 · original submission · Major Revisions

Based on the comprehensive reviews provided, it's clear that the manuscript titled “Genomic insights into CKX genes: key players in cotton fiber development and abiotic stress responses” represents a significant and valuable contribution to the field of plant genomics, particularly in understanding the role of CKX genes in cotton. The study's design, data analysis, and potential implications for plant science and biotechnology are notable. However, to ensure the manuscript meets the highest standards of scientific rigor and clarity, I recommend that the authors revise their submission to address the following major concerns:
-Functional validation of CKX genes
-a comparative analysis of CKX genes with those from other plant species could provide valuable evolutionary insights.
-The manuscript should clearly outline the rationale for selecting specific CKX genes for detailed study.
-A more thorough comparison with previous studies on CKX genes in other crops would help position the findings within the broader field of plant science, highlighting the unique contributions and implications of this research.

Reviewer 1 ·

Basic reporting

The article "Genomic insights into CKX genes: key players in cotton fiber development and abiotic stress responses" has professional article structure and is clear and unambiguous for the readers. The English language is sufficient along with the literature references.

Experimental design

The research question of the paper "Genomic insights into CKX genes: key players in cotton fiber development and abiotic stress responses" is well defined and fills an indentified gap. Methods were descibed with detail and are able to replicate.

Validity of the findings

No comment

Additional comments

In lines 83 to 95,118, it would be better to refer in third person instead of using "we".
Also in line 144 you need to specify the variety selected and also if it is GMO Variety or not.
In figure 3 and 4, please check the space between the words.

Reviewer 2 ·

Basic reporting

This manuscript is a solid piece of work on the CKX gene family in cotton, shedding light on its role in fiber growth and stress responses. The way the study is designed and the analysis is done adds something important to the plant genomics and biotech fields. But, hitting on the points mentioned would make your paper even sharper, more connected, and convincing.

Experimental design

In the section that talks about quantitative real-time PCR (qRT-PCR) analysis (pages 10-11), it'd help if the authors could throw in more details about the stats tests used to sort through expression data. Like, did ya use two-tailed tests and what was the significance level you were lookin' at (e.g., p<0.05)? This could really make your results clearer and more trustworthy.

Also, explaining why you picked the sample size for the qRT-PCR experiments would make the study seem more solid. Sayin' how you decided on the number of samples, maybe with a power analysis, could help with any doubts about your findings being strong enough.

Them heatmaps in Figure 8 (page 28) that show how GhCKX genes are expressed could do with a bit of sprucing up for clarity. You should definitely throw in a color scale or legend to make sense of the color depths. Plus, make sure the names of the genes and the conditions are easy to read; it’ll help folks understand what’s goin' on better.

The way you move from finding CKX gene family members to actually analyzing what they do could be laid out clearer in the intro or methods. It'd be great if you could spell out why certain genes got picked for a deep dive (e.g., based on how much they show up, their evolutionary importance, or what previous studies have said), to make the story you’re telling stronger.

Validity of the findings

Even though the paper does a good job looking at CKX genes in cotton, you really need to prove how these genes affect fiber growth and dealing with stress. Adding some more experiments, like turning genes off or ramping them up, to directly connect gene function with what you observe would be solid.

·

Basic reporting

.

Experimental design

.

Validity of the findings

.

Additional comments

Comments and Suggestions for Authors
The paper “Genomic insights into CKX genes: key players in cotton fiber development and abiotic stress responses” is a good research work that focuses on the explains the involvement of GhCKXs in both fibre development and response to abiotic stress.
The manuscript is well-written, and the introduction is also written with enough background information. The references (older and current) are adequate for the paper. The different sections are well organized, and the concepts are explained with good figures (figure quality is not good) and tables. The overall quality is good and worth reading by the plant science community, especially researchers working with the model plant. However, some major questions answered would further improve the manuscript.
1. How do CKX genes regulate cotton fiber development at the genomic level?
2. What genomic techniques have been employed to study CKX genes in cotton?
3. What are the specific roles of CKX genes in different stages of cotton fiber development?
4. How do CKX genes influence abiotic stress responses in cotton, and what genomic evidence supports this?
5. Can you explain the regulatory mechanisms controlling CKX gene expression in response to abiotic stressors?
6. Are there any genetic variants or polymorphisms in CKX genes associated with enhanced fiber quality or stress tolerance in cotton?
7. What are the potential applications of genomic insights into CKX genes for cotton breeding programs?
8. How do CKX genes interact with other genes or signaling pathways involved in cotton fiber development and stress responses?
9. Are there any epigenetic modifications or regulatory elements associated with CKX gene expression in cotton?
10. What challenges remain in understanding the full genomic landscape of CKX genes in cotton and their implications for fiber development and stress resilience?

Reviewer 4 ·

Basic reporting

The article titled "Genomic insights into CKX genes: key players in cotton fiber development and abiotic stress responses" presents a comprehensive study on the CKX gene family in four cotton species. It includes genome-wide identification, phylogenetic analysis, gene structure, motif analysis, and expression patterns under various conditions. The study is well-structured, employing a range of bioinformatics tools and experimental approaches to uncover the role of CKX genes in fiber development and stress responses.

Major comments
-The findings are intriguing, yet the conclusion that CKX genes significantly affect fiber elongation based on expression patterns could be strengthened by functional validation experiments, such as gene knockouts or overexpression studies in cotton.
-Some figures, particularly phylogenetic trees and gene expression heatmaps, could be improved for readability. High-resolution images and clearer labeling might help. Additionally, a comparative analysis with CKX genes from other plant species might provide deeper evolutionary insights.
-While the article cites relevant literature, a more thorough comparison with previous studies on CKX genes in other crops would contextualize the findings within the broader field of plant science.
-The study appears to meet high technical and ethical standards. However, confirming adherence to ethical guidelines for genetic studies and providing clear statements about data sharing and potential conflicts of interest would be beneficial.
In summary, the manuscript presents valuable insights into the CKX gene family in cotton, with potential implications for improving crop resilience and fiber quality.

Experimental design

no comment

Validity of the findings

no comment

---

## Round 0.2 · accepted · Accept

The authors addressed all of the reviewer's comments. This manuscript is ready for publication.